# Decoding the complexity of delayed wound healing following *Enterococcus faecalis* infection

**Cenk Celik[1†], Stella Tue Ting Lee[1], Frederick Reinhart Tanoto[2], Mark Veleba[2], Kimberly Kline[1,2,3]\*, Guillaume Thibault[1,4]\***

[1]School of Biological Sciences, Nanyang Technological University, Singapore, Singapore; [2]Singapore Centre for Environmental Life Science Engineering, Nanyang Technological University, Singapore, Singapore; [3]Department of Microbiology and Molecular Medicine, Faculty of Medicine, University of Geneva, Geneva, Switzerland; [4]Mechanobiology Institute, National University of Singapore, Singapore, Singapore

**\*For correspondence:**
kimberly.kline@unige.ch (KK);
thibault@ntu.edu.sg (GT)

**Present address:** [†]Department of Genetics, Evolution and Environment, Genetics Institute, University College London, London, United Kingdom

**Competing interest:** The authors declare that no competing interests exist.

**Abstract** Wound infections are highly prevalent and can lead to delayed or failed healing, causing significant morbidity and adverse economic impacts. These infections occur in various contexts, including diabetic foot ulcers, burns, and surgical sites. *Enterococcus faecalis* is often found in persistent non-healing wounds, but its contribution to chronic wounds remains understudied. To address this, we employed single-cell RNA sequencing (scRNA-seq) on infected wounds in comparison to uninfected wounds in a mouse model. Examining over 23,000 cells, we created a comprehensive single-cell atlas that captures the cellular and transcriptomic landscape of these wounds. Our analysis revealed unique transcriptional and metabolic alterations in infected wounds, elucidating the distinct molecular changes associated with bacterial infection compared to the normal wound healing process. We identified dysregulated keratinocyte and fibroblast transcriptomes in response to infection, jointly contributing to an anti-inflammatory environment. Notably, *E. faecalis* infection prompted a premature, incomplete epithelial-mesenchymal transition in keratinocytes. Additionally, *E. faecalis* infection modulated M2-like macrophage polarization by inhibiting pro-inflammatory resolution in vitro, in vivo, and in our scRNA-seq atlas. Furthermore, we discovered macrophage crosstalk with neutrophils, which regulates chemokine signaling pathways, while promoting anti-inflammatory interactions with endothelial cells. Overall, our findings offer new insights into the immunosuppressive role of *E. faecalis* in wound infections.

## eLife assessment

Wounds are commonly infected, which can lead to delayed or poor wound healing, thereby significantly impacting morbidity and overall quality of life for patients. This manuscript uses single cell RNA sequencing to try to understand the impact of infection on various cell types during wound healing in a mouse model. The methodology is **solid** and the results provide a **valuable** 'atlas' of the cellular changes associated with infected and uninfected wounds which will be of interest to the field.

## Introduction

Infections pose a challenge to the wound healing process, affecting both the skin's protective barrier and the efficient repair mechanisms required for tissue integrity. The skin, the largest and most intricate defense system in mammals, exhibits unique cellular heterogeneity and complexity

**eLife digest** If wounds get infected, they heal much more slowly, sometimes leading to skin damage and other complications, including disseminated infections or even amputation. Infections can happen in many types of wounds, ranging from ulcers in patients with diabetes to severe burns. If infections are not cleared quickly, the wounds can become 'chronic' and are unable to heal without intervention.

*Enterococcus faecalis* is a type of bacteria that normally lives in the gut. Within that environment, in healthy people, it is not harmful. However, if it comes into contact with wounds – particularly diabetic ulcers or the site of a surgery – it can cause persistent infections and prevent healing.

Although researchers are beginning to understand how *E. faecalis* initially colonises wounds, the biological mechanisms that transform these infections into chronic wounds are still largely unknown. Celik et al. therefore set out to investigate exactly how *E. faecalis* interferes with wound healing.

To do this, Celik et al. looked at *E. faecalis*-infected wounds in mice and compared them to uninfected ones. Using a genetic technique called single-cell RNA sequencing, Celik et al. were able to determine which genes were switched on in individual skin and immune cells at the site of the wounds. This in turn allowed the researchers to determine how those cells were behaving in both infected and uninfected conditions.

The experiments revealed that when *E. faecalis* was present in wounds, several important cell types in the wounds did not behave normally. For example, although the infected skin cells still underwent a change in behaviour required for healing (called an epithelial-mesenchymal transition), the change was both premature and incomplete. In other words, the skin cells in infected wounds started changing too early and did not finish the healing process properly.

*E. faecalis* also changed the way macrophages and neutrophils worked within the wounds. These are cells in our immune system that normally promote inflammation, a process involved in both uninfected wounds or during infections and is a key part of wound healing when properly controlled. In the *E. faecalis*-infected wounds, these cells' inflammatory properties were suppressed, making them less helpful for healing.

These results shed new light on how *E. faecalis* interacts with skin cells and the immune system to disrupt wound healing. Celik et al. hope that this knowledge will allow us to find new ways to target *E. faecalis* infections, and ultimately develop treatments to help chronic wounds heal better and faster.

that maintain tissue homeostasis. Within the skin, undifferentiated resident cells are the main modulators of tissue maintenance, albeit terminal differentiation dynamics adapt to the regenerative requirements of the tissue. Several comprehensive studies conducted in mice and humans have highlighted the complexity of the skin (*Cheng et al., 2018*; *Der et al., 2019*; *Joost et al., 2020*; *Joost et al., 2018*; *Joost et al., 2016*; *Philippeos et al., 2018*; *Theocharidis et al., 2022*). However, these studies have primarily focused on cellular heterogeneity and the transcriptome of intact skin or uninfected wounds, offering limited insights into the dynamics of wound healing following infection.

Efficient healing during wound infections involves a sequence of events occurring in three distinct but interconnected stages: inflammation, proliferation, and remodeling (*Eming et al., 2014*; *Masson-Meyers et al., 2020*; *Minutti et al., 2017*; *Rognoni and Watt, 2018*). These events involve a series of inter- and intracellular molecular interactions mediated by soluble ligands and the innate immune system (*Lindley et al., 2016*; *Wang et al., 2018*). The inflammatory phase initiates migration of leukocytes into the wound site to help clear cell debris and establish tissue protection through local inflammation. Initially, neutrophils, responding to pro-inflammatory cytokines like IL-1β, TNF-α, and IFN-γ, extravasate through the endothelium to the site of injury. Subsequently, the proliferative stage aims to reduce wound size through contraction and re-establishment of the epithelial barrier. This stage is facilitated by activated keratinocytes modulated by inflammatory responses, cytokines, and growth factors. In the final stage, tissue remodeling restores the mechanical properties of intact skin through ECM reorganization, degradation, and synthesis (*Wang et al., 2018*). Dysregulation of this intricate mechanism can lead to pathological outcomes characterized by persistent inflammation and excessive ECM production (*Ashcroft et al., 2013*).

*Enterococcus faecalis* is a commensal bacterium in the human gut, and also an opportunistic pathogen responsible for various infections, including surgical site infections and diabetic ulcers (*Kao and Kline, 2019*). Bacterial wound infections in general, including those associated with *E. faecalis*, are biofilm-associated, resulting in an antibiotic-tolerant population that may lead to persistent infections (*Ch'ng et al., 2019*). Additionally, *E. faecalis* has mechanisms to evade and suppress immune clearance by, for example, suppressing the pro-inflammatory M1-like phenotype in macrophage and preventing neutrophil extracellular trap formation (*Kao et al., 2023*; *Kao and Kline, 2019*; *Tien et al., 2017*). *E. faecalis* can also persist (*Bertuccini et al., 2002*; *Gentry-Weeks et al., 1999*; *Horsley et al., 2018*; *Horsley et al., 2013*; *Olmsted et al., 1994*; *Wells et al., 1990*; *Wells et al., 1988*; *Zou and Shankar, 2014*; *Zou and Shankar, 2016*) and replicate (*da Silva et al., 2022*; *Nunez et al., 2022*) within epithelial cells and macrophages, further complicating treatment. The combination of biofilm formation and immune evasion makes Enterococcal wound infections a significant clinical challenge.

In a murine full-thickness excisional wound model, our prior investigations revealed a bifurcated trajectory characterizing *E. faecalis* wound infections (*Chong et al., 2017*). Initially, bacterial colony-forming units (CFU) increase in number in the acute replication phase, with a concomitant pro-inflammatory response characterized by pro-inflammatory cytokine and chemokine production coupled with neutrophil infiltration. In the subsequent persistence stage, *E. faecalis* CFU undergo gradual reduction and stabilization at approximately $10^5$ CFU within wounds by 2–3 day post-infection (dpi), coinciding with delayed wound healing. Within this framework, we have identified specific bacterial determinants that contribute to each phase of *E. faecalis* infection. For example, de novo purine biosynthesis emerges as a pivotal factor in enhancing bacterial fitness during the acute replication phase (*Tan et al., 2022*). In the persistent phase at 3 dpi, the galactose and mannose uptake systems, in conjunction with the *mprF* gene product, are associated with nutrient acquisition and resistance to antimicrobial peptides and neutrophil-mediated killing, respectively (*Bao et al., 2012*; *Chong et al., 2017*; *Jin et al., 2021*; *Kandaswamy et al., 2013*; *Rashid et al., 2023*). Nonetheless, the intricate interplay between these *E. faecalis* persistence-associated virulence factors, their immunomodulatory evasion strategies, and precise implications in the context of delayed wound healing remains to be investigated.

To address this, we generated a comprehensive single-cell atlas of the host response to persistent *E. faecalis* wound infection. We observed that *E. faecalis* induces immunosuppression in keratinocytes and fibroblasts, delaying the immune response. Notably, *E. faecalis* infection prompted a partial epithelial-mesenchymal transition (EMT) in keratinocytes. Moreover, macrophages in infected wounds displayed M2-like polarization. Our findings also indicate that the interactions between macrophages and endothelial cells contribute to the anti-inflammatory niche during infection. Furthermore, *E. faecalis*-infected macrophages drive pathogenic vascularization signatures in endothelial cells, resembling the tumor microenvironment in cancer. We also noted *E. faecalis* infected-associated macrophage crosstalk with neutrophils, regulating chemokine signaling pathways and promoting anti-inflammatory interactions with endothelial cells. These insights from our scRNA-seq atlas provide a foundation for future studies aimed at investigating bacterial factors contributing to wound pathogenesis and understanding the underlying mechanisms associated with delayed healing.

## Results

### *E. faecalis* infection inhibits wound healing signatures

In the wound environment, various cell types, including myeloid cells, fibroblasts, and endothelial cells, play critical roles in the initial and later stages of wound healing by releasing platelet-derived growth factor (PDGF). Additionally, macrophages secrete epidermal growth factor (EGF) in injured skin, which operates through the epidermal growth factor receptor (EGFR) to promote keratinocyte proliferation, migration, and re-epithelialization. To understand the impact of *E. faecalis* infection on wound healing, we infected excisional dorsal wounds on C57BL/6J male mice and measured gene expression levels of wound healing markers in bulk tissue at 4 dpi at the onset of persistent infection. We then compared the expression profiles with those of (i) wounded but uninfected or (ii) unwounded and uninfected controls, ensuring that the uninfected and infected mice were of comparable weight (*Figure 1A*, *Figure 1—figure supplement 1A, B*). We observed lower expression of platelet-derived growth factor subunit A (*Pdgfa*) in *E. faecalis*-infected wounds than in unwounded

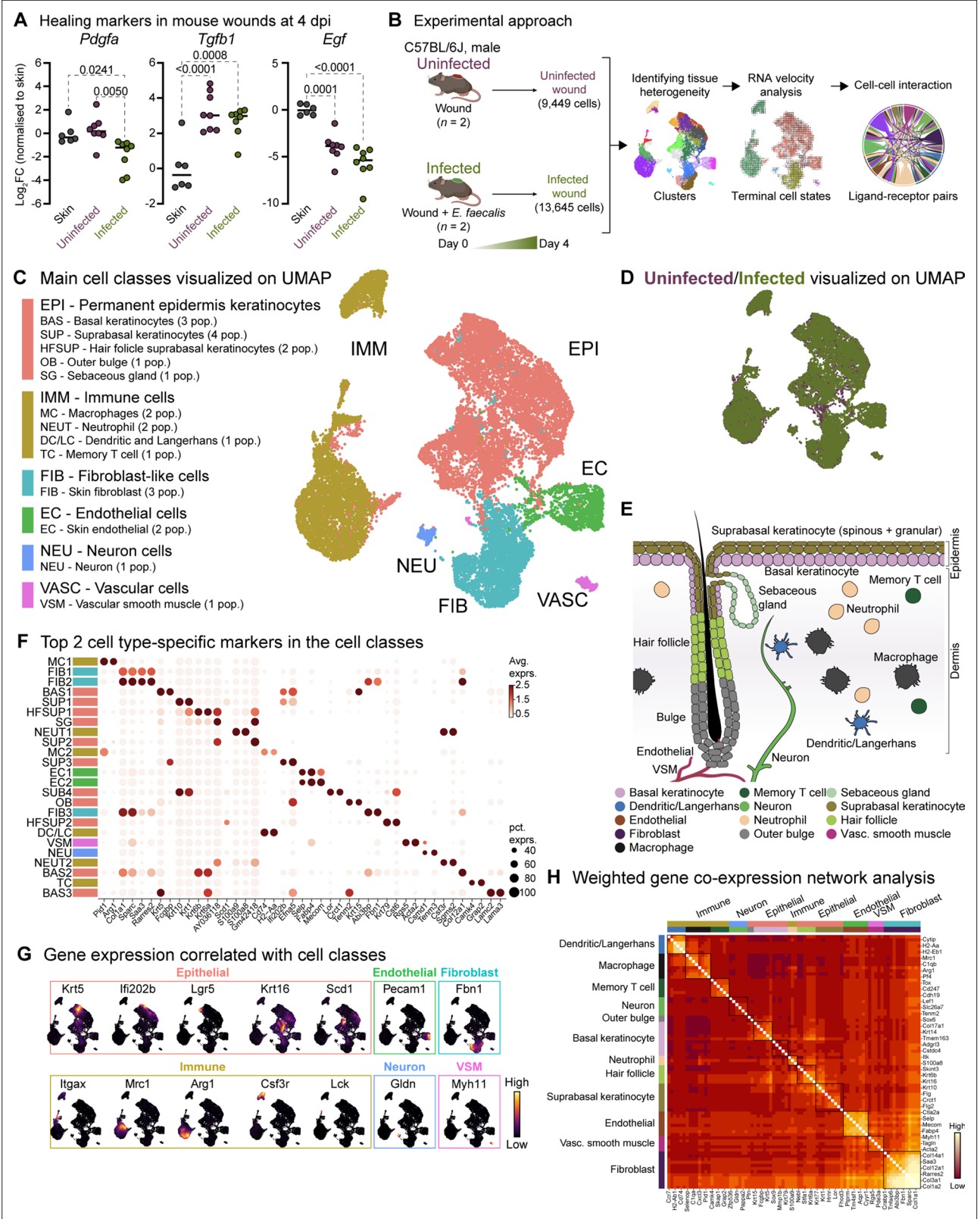

**Figure 1.** Mouse skin wound infection atlas. (**A**) Gene expression of healing markers *Pdgfa*, *Tgfb1,* and *Egf* 4 days post-infection (4 dpi) for uninfected and *E. faecalis*-infected 6–7 week-old C57BL/6 J mouse skin wounds, normalized to intact skin (n=6 for skin; n=8 for wounds; one-way ANOVA). (**B**) Single-cell RNA sequencing workflow of full-thickness mouse wounds. (**C**) Integrated dataset of ~23,000 scRNA-seq libraries from uninfected and infected wounds identifies 5 mega cell classes indicated in the UMAP. (**D**) UMAP colored by the uninfected and infected conditions. (**E**) Schematic

*Figure 1 continued on next page*

Figure 1 continued

describing the color-matched cell types with that of the clusters in Figure S1F. (**F**) Dot plot depicting the top two cell type-specific markers in the integrated data. Legend indicates average expression and dot size represents percent expression. (**G**) Density plots depict cell types described in C. (**H**) Heat map of weighted gene co-expression network analysis for annotated cell populations, colored by bars matching mega clusters and annotated cell types in C.

The online version of this article includes the following figure supplement(s) for figure 1:

**Figure supplement 1.** Comparison of 4dpi characteristics between uninfected and *E. faecalis* infected in vivo and scRNA-seq data.

skin, while uninfected wounds remained unchanged. Similarly, the expression of *Egf* was lower in uninfected wounds than in healthy skin, while reaching the lowest levels in *E. faecalis*-infected wounds. The reduced *Egf* expression in uninfected wounds at 4 days post-wounding is expected, as EGF primarily influences skin cell growth, proliferation, and differentiation during the later stages of wound healing, typically around 10 days post-injury (*Schultz et al., 1991*). By contrast, wounding alone resulted in higher transforming growth factor beta 1 (*Tgfb1*) expression. TGF-β1 plays a dual role in wound healing depending on the microenvironment. During tissue maintenance, TGF-β1 acts as a growth inhibitor, and its absence in the epidermis leads to keratinocyte hyperproliferation (*Guasch et al., 2007*). Elevated TGF-β1 levels have also been observed in the epidermis of chronic wounds in both humans and mouse models (*Li et al., 2021*; *Xie et al., 2009*). In addition, we observed a slightly higher expression of fibroblast growth factor 1 (*Fgf1*) in uninfected wounds, although the differences were comparable (*Figure 1—figure supplement 1A*). Overall, the persistent *E. faecalis* infection contributed to higher *Tgfb1* expression, whilst *Pdgfa* levels remained low, correlating with delayed wound healing.

## Single-cell transcriptomes of full-thickness skin wounds diverge during healing and infection

To understand the cellular heterogeneity in the wound environment following *E. faecalis* infection, we dissociated cells from full-thickness wounds, including minimal adjacent healthy skin. We generated single-cell transcriptome libraries using the droplet-based 10 X Genomics Chromium microfluidic partitioning system (*Figure 1B* and *Supplementary file 1*). By integrating the transcriptomes of approximately 23,000 cells, we employed the unbiased graph-based Louvain algorithm to identify clusters (*Zappia and Oshlack, 2018*). Our analysis revealed 24 clusters, irrespective of infection (*Figure 1—figure supplement 1C*). These clusters included epithelial, immune, fibroblast, endothelial, vascular, and neural cell types (*Figure 1C and D*, *Figure 1—figure supplement 1D and E*, and *Supplementary files 2-4*). Within the epithelial class, we identified basal (BAS1-3) and suprabasal (SUP1-4) keratinocytes, hair follicles (HFSUP1-2), outer bulge (OB), and sebaceous gland (SG) cells (*Figure 1E*). The immune class included macrophages (MC1-2), neutrophils (NEUT1-2), memory T cells (TC), and a mixed population (DC/LC) of dendritic cells (DCs) and Langerhans cells (LC). Notably, cells from infected wounds (~13,000 cells) demonstrated distinct clustering patterns compared to cells from uninfected wounds (~9500 cells; *Figure 1—figure supplement 1F* and *Supplementary file 1*).

We proceeded to analyze upregulated gene signatures for each Louvain cluster (*Figure 1F* and *Supplementary file 2*) and compared them to the cell-type signature database (*Franzén et al., 2019*; PanglaoDB) to identify characteristic gene expression patterns (*Figure 1G*). Additionally, we identified highly expressed genes in uninfected and infected wounds (*Supplementary files 3 and 4*) and performed co-expression network analysis (WGCNA) to uncover the core genes associated with cell types (*Figure 1H*).

We also delineated macrophage populations (*Figure 1—figure supplement 1G–I*), identifying an M2-like polarization (*Figure 1F*, MC1) marked by higher expression of *Arg1*, *Egr2*, *Fn1*, and *Fpr2* (*Figure 1—figure supplement 1G*), and a tissue-resident macrophage (TRM) population (*Figure 1F*, MC2) expressing *Ccr5*, *Cd68*, *Fcgr1*, *Mrc1* (*Cd206*), *Ms4a4c*, and *Pparg* (*Figure 1—figure supplement 1H*). Both macrophage populations exhibited limited expression of *Grp18* and *Nos2* (*iNos*), which are typically associated with M1-like polarization (*Figure 1—figure supplement 1I*). Together, our analysis outlined the differences observed in uninfected and infected wound healing across epithelial, fibroblast, and immune cell populations.

## *E. faecalis* elicit immunosuppressive interactions with keratinocytes

Keratinocytes, the predominant cell population in the skin, include undifferentiated (basal) and differentiated (suprabasal, hair follicle, outer bulge, and sebaceous gland) cells within the EPI cell class in both wound types (*Figure 1C*). Further analysis of the EPI class revealed 19 clusters (*Figure 2A* and *Supplementary file 5*), with the emergence of infection-specific clusters (*Figure 2B*, clusters 0 and 6, and *Figure 2—figure supplement 1A*). We identified three major keratinocyte populations: (i) *Krt5*-expressing (*Krt5^{hi}*) basal keratinocytes, (ii) *Krt10*-enriched (*Krt10^{hi}*) post-mitotic keratinocytes, and (iii) *Krt5^{hi}Krt10^{hi}* co-expressing populations (*Figure 2C*). The co-expression of *Krt5* and *Krt10* was unique to infection-specific clusters (*Figure 2A*, clusters 0 and 6). We also observed distinct populations, including hair follicle stem cells (*Lrg5^{hi}*; cluster 10), differentiating basal cells (*Ivl^{hi}*; clusters 7 and 8), terminally differentiated keratinocytes (*Lor^{hi}*; clusters 2, 8, 16, 17), and bulge stem cells (*Krt15^{hi}*; clusters 2, 9, 10, 17, 18; *Figure 2D*), reflecting the skin's complexity.

The two largest infection-specific clusters exhibited distinct gene expression patterns, with cluster 0 and cluster 6, enriched in *Zeb2* and *Gjb6* expression, respectively (*Figure 2A*). These clusters represent migratory stem-like and partial EMT keratinocyte populations (*Figure 2E*). As the barrier to pathogens, keratinocytes secrete a broad range of cytokines that can induce inflammatory responses (*Alshetaiwi et al., 2020*; *Siriwach et al., 2022*; *Veglia et al., 2021*). However, *Zeb2^{hi}* keratinocytes co-expressing *Cxcl2*, *Il1b*, and *Wfdc17*, indicate myeloid-derived suppressor cell-like phenotype which implies an immunosuppressive environment (*Hofer et al., 2021*; *Veglia et al., 2021*). Gene Ontology analysis revealed that the *Zeb2^{hi}* and *Gjb6^{hi}* clusters exhibited signatures related to ECM remodeling, chemokine signaling, migratory pathways, and inflammatory response (*Figure 2F and G*, and *Supplementary file 5*). Collectively, these findings suggest an early migratory role of keratinocytes induced by *E. faecalis* infection.

During cutaneous wound healing, keratinocytes enter a mesenchymal-like state, migrating to and proliferating within the wound site. We observed two infection-specific keratinocyte populations enriched in *Zeb2* and *Gjb6* expression, respectively, as early as 4 days post-wounding. To determine the nature of these populations, we performed RNA velocity analysis, a method that predicts the future state of individual cells based on the patterns of temporal gene expression (*La Manno et al., 2018*). RNA velocity analysis predicted a lineage relationship between *Zeb2^{hi}* and *Gjb6^{hi}* keratinocytes (*Figure 2H and I*, and *Supplementary file 5*). The top lineage-driver genes (*Lange et al., 2022*), including *Rgs1*, *H2-Aa*, *Ms4a6c*, *Cd74*, *H2-Eb1*, and *H2-Ab1*, were predominantly expressed in the infection-specific cluster 0 (*Figure 2J* and *Figure 2—figure supplement 1B*). These genes are associated with the major histocompatibility complex (MHC) class II, suggesting a self-antigen presenting keratinocyte population, which have a role in co-stimulation of T cell responses (*Meister et al., 2015*; *Tamoutounour et al., 2019*). Meanwhile, *Gjb6^{hi}* keratinocytes demonstrated reduced expression of putative genes such as *Pof1b*, *Krt77*, *Dnase1l3*, and *Krtdap* as well as increased *Clic4* expression (*Figure 2—figure supplement 1C*), suggesting that *E. faecalis* infection perturbs normal healing. Additionally, temporal expression analysis of high-likelihood genes revealed three distinct transcriptional states (*Figure 2K*): (1) an early state characterized by undifferentiated (basal) keratinocyte markers, (2) an intermediate state defined by the selection and upkeep of intraepithelial T cell protein family, and (3) a late state characterized by cell adhesion signatures. These findings provide insights into the cellular dynamics and developmental abnormalities, such as partial EMT induced by *E. faecalis* infection during wound healing.

Understanding cell-cell crosstalk through ligand-receptor interactions allows predicting ligand-target links between interacting cells by combining their expression data with signaling and gene regulatory network databases. Given our observation that infection-specific keratinocyte populations (*Figure 2—figure supplement 1*) were involved in ECM remodeling and immune response (*Figure 2F and G*), we hypothesized that these cells might participate in the SPP1 signaling pathway. SPP1 (secreted phosphoprotein 1 or osteopontin) is a chemokine-like protein secreted by immune cells, osteoblasts, osteocytes, and epithelial cells to facilitate anti-apoptotic immune responses (*Denhardt et al., 2001*; *Standal et al., 2004*). To decipher ligand-receptor interactions in uninfected and infected skin wounds, we performed cell-cell interaction analysis (*Guerrero-Juarez et al., 2019*). We found 34 predicted interactions in uninfected keratinocytes and 61 in keratinocytes from *E. faecalis*-infected wounds out of a total of 923 and 991 interactions, respectively, in our single-cell atlas (*Figure 1—figure supplement 1A and B*, and *Supplementary file 6*). Importantly, we detected outgoing signals

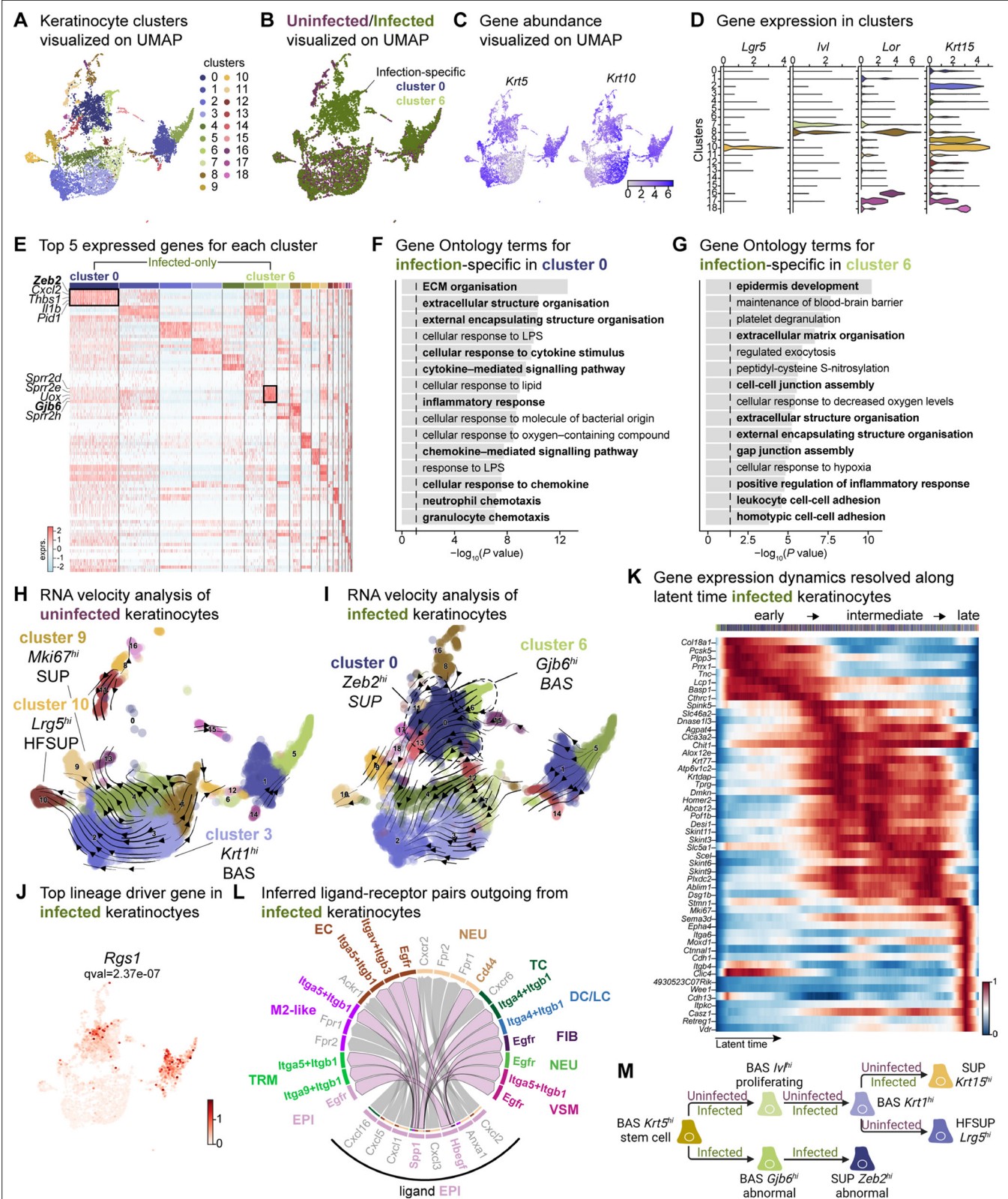

**Figure 2.** Sub-clustering of keratinocyte populations reveals infection-specific cell types. (**A**) UMAP of integrated keratinocyte (basal, suprabasal, hair follicle, bulge, and sebaceous gland) population reveals 19 clusters. (**B**) Infected keratinocytes (green) show unique and shared clusters with uninfected keratinocytes (purple). (**C**) Spatial dispersion of *Krt5* and *Krt10* abundance in keratinocytes. (**D**) Expression of *Lgr5*, *Ivl*, *Lor*, and *Krt15* in Louvain clusters shown in A. (**E**) Heat map of top 5 differentially expressed marker genes for each cluster in keratinocytes. Rectangle boxes indicate infection-

*Figure 2 continued on next page*

*Figure 2 continued*

specific Louvain clusters. (**F–G**) The bar plots show the top 15 Gene Ontology terms for infection-specific (**F**) *Zeb2*[hi] (cluster 0) and (**G**) *Gjb6*[hi] (cluster 6) keratinocyte populations. (**H–I**) Dynamic RNA velocity estimation of uninfected (**H**) and infected (**I**) keratinocytes. (**J**) The top lineage driver gene, *Rgs1*, in infected keratinocytes, was ubiquitously expressed in infection-specific Louvain clusters. (**K**) Gene expression dynamics resolved along latent time in the top 50 likelihood-ranked genes of infected keratinocytes. The colored bar at the top indicates Louvain clusters in I. Legend describes scaled gene expression. (**L**) Inferred ligand-receptor pairs outgoing from infected keratinocytes. (**M**) The hierarchy tree depicts the trajectory of differentiating and terminally differentiated keratinocyte cells originating from basal keratinocytes. Panel M created with BioRender.com, and published using a CC BY-NC-ND license with permission.

The online version of this article includes the following figure supplement(s) for figure 2:

**Figure supplement 1.** Infection-specific keratinocytes display differential transcriptome.

**Figure supplement 2.** Overall cellular interactome of scRNA-seq atlas.

from keratinocytes to other cells associated with EGF and SPP1 signaling pathways. Ligand:receptor pairs in the infected niche (*Figure 2—figure supplement 1D*) included Hbegf:Egfr for the EGF pathway and Spp1:Cd44, Spp1:(Itga5 +Itgb1), Spp1:(Itga9 +Itgb1), and Spp1:(Itgav +Itgb3) for the SPP1 pathway (*Figure 2L*, *Figure 2—figure supplement 1D and E*), that are known to induce immunosuppression (*Cheng et al., 2023*; *Gao et al., 2022*). Remarkably, we observed the enrichment of keratinocyte-endothelial cell ligand:receptor pairs (*Figure 2—figure supplement 2* and *Supplementary file 6*). By contrast, keratinocytes from uninfected wounds only showed macrophage migration inhibitory factor (Mif) ligand interactions with its receptors Cd44, Cd74, and Cxcr4 in immune cells (DC/LC and TRM) (*Figure 2—figure supplement 1G*), suggesting that these keratinocytes promote cell proliferation, wound healing, and survival (*Farr et al., 2020*; *Jäger et al., 2020*). Furthermore, the RNA velocity of keratinocytes from uninfected wounds revealed a terminal hair follicle (*Lrg5*[hi]) and a proliferating keratinocyte (*Mki67*[hi]) population originating from basal keratinocytes (*Figure 2H*), underlining normal wound healing. Overall, *E. faecalis* infection altered the transcriptome of keratinocytes toward a partial EMT at an early stage, whereas uninfected keratinocytes showed differentiating and terminally differentiated keratinocyte populations (*Figure 2M*). Our data show that epidermal cells undergo migratory and inflammatory gene regulation during normal wound healing (*Haensel et al., 2020*; *Vu et al., 2022*), whereas *E. faecalis* induces anti-inflammatory transcriptional modulation that may promote chronicity in bacteria-infected skin wounds. These findings demonstrate that keratinocytes exist in a low inflammation profile under homeostasis, which is exacerbated upon *E. faecalis* infection.

## *E. faecalis* delays immune response in fibroblasts

Fibroblasts are the fundamental connective tissue cells involved in skin homeostasis and healing. Upon injury, they primarily (1) migrate into the wound site, (2) produce ECM by secreting growth factors, and (3) regulate the inflammatory response. Despite their high heterogeneity, fibroblast subpopulations have distinct roles in wound healing. In response to injury, fibroblasts produce increased amounts of ECM by inhibiting the metalloproteinase (MMP) family through the tissue inhibitor of metalloproteinases 1 (TIMP1). Notably, *Guerrero-Juarez et al., 2019* reported a rare fibroblast population expressing myeloid lineage cell markers during normal (uninfected) wound healing (*Guerrero-Juarez et al., 2019*). Our bioinformatic analysis identified fibroblast populations within wounds that differed significantly between infected and uninfected conditions (*Figure 1—figure supplement 1F*). Further analysis of the fibroblast mega class revealed 12 distinct clusters, with clusters 0 and 2 specific to infection while retaining their fibroblastic identity (*Figure 3A and B*, and *Supplementary file 7*). The infection-associated fibroblasts express a range of markers, including those linked to extracellular matrix deposition such as *Col1a1*, *Col1a2*, *Col6a2*, vimentin (*Vim*), fibronectin, elastin (*Eln*), asporin (*Aspn*), platelet-derived growth factor receptor-beta (*Pdgfrb*), and fibroblast activator protein-α (*Fap*). These findings suggest a premature extracellular matrix deposition triggered by *E. faecalis*, which typically occurs in later stages of wound healing processes (*Deng et al., 2021*; *Fang et al., 2023*; *Fitzgerald and Weiner, 2020*; *Figure 3C and D*, *Figure 3—figure supplement 1A and B*).

The unique fibroblast clusters 0 and 2 associated with *E. faecalis* infection (*Figure 3E*) exhibited enrichment of myeloid-specific markers (*Hbb-bs*, *Lyz2*) and profibrotic gene signatures (*Inhba*, *Saa3*,

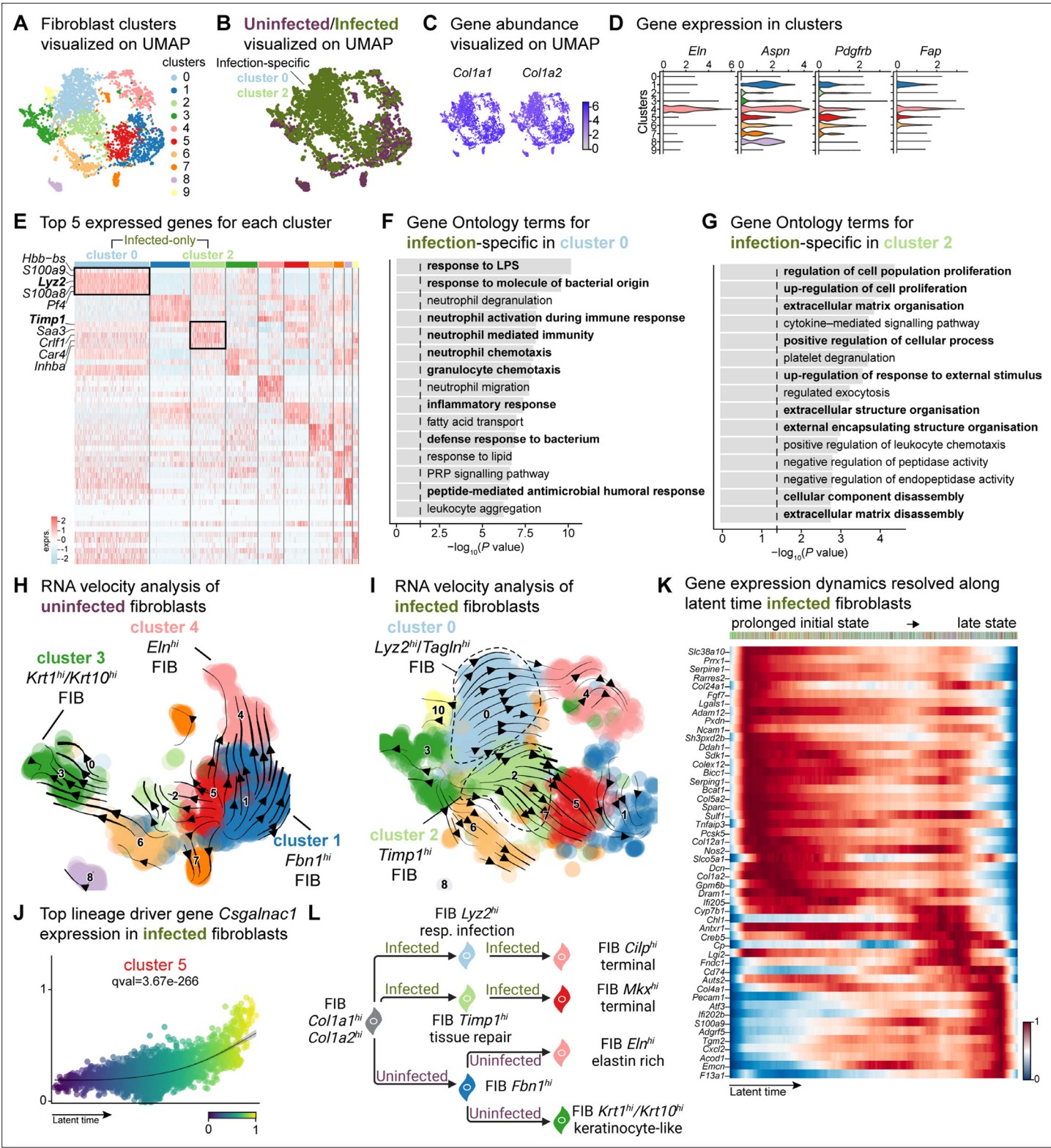

**Figure 3.** *E. faecalis* delays immune response in fibroblasts. (**A**) UMAP of integrated fibroblasts reveals 12 clusters. (**B**) Infected fibroblasts (green) show unique and shared clusters with uninfected fibroblasts (purple). (**C**) Spatial dispersion of *Col1a1* and *Col1a2* abundance in fibroblasts. (**D**) Expression of *Eln*, *Aspn*, *Pdgfrb*, and *Fap* in Louvain clusters shown in A. (**E**) Heat map of top 5 differentially expressed marker genes for each cluster in fibroblasts. Rectangle boxes indicate infection-specific Louvain clusters. (**F–G**) The bar plots show the top 15 Gene Ontology terms for infection-specific (**F**) *Lyz2^hi^/Tagln^hi^* (cluster 0) and (**G**) *Timp1^hi^* (cluster 2) fibroblast populations. (**H–I**) Dynamic RNA velocity estimation of uninfected (**H**) and infected

Figure 3 continued

(I) fibroblasts. (J) The top lineage driver gene, *Csgalnac1*, expression in infected fibroblasts. (K) Gene expression dynamics resolved along latent time in the top 50 likelihood-ranked genes of infected fibroblasts. The colored bar at the top indicates Louvain clusters in I. Legend describes scaled gene expression. (L) While uninfected fibroblasts show healing phenotypes, infected fibroblasts undergo two transitioning phases: (i) contractile and (ii) pathologic. Panel L created with BioRender.com, and published using a CC BY-NC-ND license with permission.

The online version of this article includes the following figure supplement(s) for figure 3:

**Figure supplement 1.** Extended analysis of the fibroblast population reveals unique signatures.

*Timp1*; *Supplementary file 6*). Additionally, *Lyz2*$^{hi}$ fibroblasts co-expressing *Tagln*, *Acta2*, and *Col12a1* suggested their origin as contractile myofibroblasts (*Guerrero-Juarez et al., 2019*). Gene Ontology analysis indicated that *Lyz2*$^{hi}$ fibroblasts were involved in immune response (*Figure 3F*), while *Timp1*$^{hi}$ fibroblasts were associated with tissue repair processes (*Figure 3G*). These findings collectively reveal an intricate cellular landscape within infected wounds, highlighting the detrimental functional contributions of various fibroblast subpopulations in response to *E. faecalis* infection.

To validate the distinct fibroblast cell states and their differentiation potential, we performed RNA velocity analysis (*Figure 3H and I*), which revealed two terminal fibroblast populations: (1) a *Cilp*$^{hi}$ fibrotic cluster 4 and (2) an *Mkx*$^{hi}$ reparative cluster 5, originating from clusters 0 and 2 (*Figure 3I* and *Supplementary file 7*), indicating the mesenchymal characteristics of infection-specific fibroblasts. Lineage driver gene analysis supported these findings, with the expression of genes such as *Csgalnact1*, *Atrnl1*, *Slit3*, *Rbms3*, and *Magi2*, which are known to induce fibrotic tissue formation under pathological conditions (*Górnicki et al., 2022*; *Mizumoto et al., 2020*; *Figure 3J* and *Figure 3—figure supplement 1C*). Similarly, the activation of genes such as *Serping1*, *Sparc*, *Pcsk5*, *Fgf7*, and *Cyp7b1* (*Figure 3—figure supplement 1D*) indicated the temporal activation of cell migration and immune suppression during infection. Cell Rank analysis revealed two states: an initial prolonged state and a short terminal state (*Figure 3K*). The early transcriptional states were associated with apoptosis inhibition, cell adhesion, and immune evasion genes, including serine protease inhibitors (*Serpine1* and *Serping1*), *Prrx1*, *Ncam1*, and *Chl1*, suggesting immune suppression. By contrast, the terminal state showed the emergence of migratory (*Cd74*, *Ifi202b*) and inflammatory (*Acod1*, *Cxcl2*, *F13a1*, and *S100a9*) genes, indicating a delayed immune response to *E. faecalis* infection. Overall, these findings indicate that uninfected fibroblasts contribute to healthy wound healing with proliferative and elastin-rich fibroblast populations. By contrast, *E. faecalis*-infected fibroblasts exhibit a pathological repair profile, characterized by the presence of *Timp1*$^{hi}$ and *Lyz2*$^{hi}$ fibroblasts (*Figure 3L*).

To understand the role of fibroblast subpopulations in healing, we explored cell-cell interactions in fibroblasts in infected wound (*Figure 3—figure supplement 1E*). In *E. faecalis*-infected wounds, we observed interactions involving Spp1 ligand with cell adhesion receptor Cd44 and integrins (Itgav+Itgb1, Itgav+Itgb3, Itgav+Itgb5, Itga4+Itgb1, Itga5+Itgb1, Itga9+Itgb1), and EGF signaling pairing Ereg:Egfr (*Figure 3—figure supplement 1F and G*). These interactions were particularly strong between macrophages (ligands) and endothelial cells (receptors; *Figure 2—figure supplement 1C*), suggesting their involvement in the infected wound microenvironment. By contrast, uninfected wounds showed a normal wound healing profile with reparative VEGF and TGF-β signaling pathways (*Figure 3—figure supplement 1H and I*). Furthermore, RNA velocity analysis of fibroblasts from uninfected wounds revealed two terminal fibroblast populations originating from *Fbn1*$^{hi}$ fibroblasts: elastin-rich (*Eln*$^{hi}$) fibroblasts and keratinocyte-like (*Krt1*$^{hi}$/*Krt10*$^{hi}$) fibroblasts (*Figure 3H*). Collectively, our findings revealed the unique roles of fibroblasts upon *E. faecalis* infection and reaffirmed the known functions of fibroblasts during uninfected wound healing, consistent with the outcomes of predicted ligand-receptor interactions (*Figure 3F*).

## *E. faecalis* promotes macrophage polarization toward an anti-inflammatory phenotype

Immune cells play a crucial role in wound healing by eliminating pathogens, promoting keratinocyte and fibroblast activity, and resolving inflammation and tissue repair (*Haensel et al., 2020*; *Landén et al., 2016*; *Vu et al., 2022*). Our data reveal unique clusters enriched within keratinocytes and

fibroblasts in *E. faecalis*-infected wounds, suggesting an immunosuppressive transcriptional program in these cell populations (*Figure 2—figure supplement 1A* and *Figure 3—figure supplement 1B*). Given the ability of *E. faecalis* to actively suppress macrophage activation in vitro (*Tien et al., 2017*), we investigated if *E. faecalis* contributes to anti-inflammatory macrophage polarization and immune suppression in vivo. Analysis of the myeloid cells identified two infection-specific macrophage clusters (clusters 2 and 5) among 12 clusters (*Figure 4A and B*, and *Supplementary file 8*). To validate infection-specific macrophage polarization, we performed qPCR on bulk tissue isolated from uninfected and infected wounds at 4 dpi. Infected wounds showed downregulation of *Mrc1* and upregulation of both *Arg1* and *Nos2* compared to uninfected wounds or unwounded bulk skin tissue (*Figure 4D*), which was further corroborated by examining gene expression in bone-marrow-derived macrophages (BMDM) following *E. faecalis* infection in vitro (*Figure 4E*). The co-expression of both M1-like and M2-like macrophage markers during infection has been previously reported for *M. tuberculosis* (*Mattila et al., 2013*), *T. cruzi* (*Cuervo et al., 2011*), and *G. lamblia* infections (*Maloney et al., 2015*), as an indicator of an immunosuppressive phenotype that promotes an anti-inflammatory environment during prolonged infection. In myeloid clusters of our scRNA-seq atlas, tissue-resident macrophages (*Mrc1*) were predominant in uninfected wounds, while M2-like macrophages (*Arg1*) were abundant in macrophages from *E. faecalis*-infected wounds (*Figure 4F and G*), supporting the hypothesis of an anti-inflammatory wound environment during *E. faecalis* infection. Additionally, our analysis allowed the segregation of dendritic cells and Langerhans cells (cluster 10, *Cd207* and *Itgax*) from macrophage populations, characterized by the co-expression of langerin (*Cd207*) and integrin alpha X [*Itgax* (*Cd11c*)] in cluster 10 (*Figure 4G*), confirming the validity of our cell annotation.

Next, we focused on the functions of infection-specific macrophage clusters 2 and 5 (*Figure 4B* and *Figure 4—figure supplement 1B*) to identify their potential roles in wound infection. Gene Ontology analysis revealed that these macrophages were involved in the immune response, ECM production, and protein translation (*Figure 4—figure supplement 1C and D*). To better understand the evolution of infection-specific macrophages, we computed the RNA velocity to explore potential lineage relationships (*Figure 4H and I*). RNA velocity identified cluster 2 macrophages as the terminal population in the infected dataset (*Figure 4I*), whereas uninfected terminal macrophage populations were found as clusters 1 and 4 (*Figure 4H*). As expected, cluster 5 did not exhibit velocity vectors since these cells were highly segregated from the main myeloid clusters and differed vastly in their gene expression (*Supplementary file 8*). While both terminal macrophage populations in both uninfected and infection conditions co-host anti-inflammatory macrophage signatures, their transcriptional program and functions vary in the presence of *E. faecalis* infection.

To further characterize clusters 2 and 5, we explored differentially expressed genes in infected cells. We identified the inflammation-related genes *Fth1*, *Slpi*, *Il1b*, *Nmes1* [*AA467197* (miR-147)], *Ptges,* and *Cxcl3* (*Figure 4J* and *Figure 4—figure supplement 1E*), suggesting that these cells may play a role in tissue remodeling (*Munadziroh et al., 2022*; *Recalcati et al., 2019*). Similarly, the expression of the infected-specific top likelihood genes *Cxcl2*, *Cd36*, and *Pdpn* was associated with the presence of efferocytic M2-like macrophages (*Figure 4—figure supplement 1F*, green clusters). The distant *Sparc*$^{hi}$ macrophage cluster 5 exhibited C-C chemokine receptor type 7 (*Ccr7*) exhaustion (*Figure 4—figure supplement 1F*, red clusters), indicating that these cells might be of M1-like origin (*Hu et al., 2020*).

Next, to validate whether the terminal macrophage populations corresponding to clusters 2 and 5 were tissue-resident or M2-like macrophages, we computed the driver genes of macrophage subclusters over latent time. The top 50 likelihood gene heat map identified two major states in the macrophages (*Figure 4K*). The early state cells were enriched in antigen-presenting/processing MHC class II gene expression such as *H2-Ab1*, *H2-Eb1*, *H2-Eb2,* G-protein-coupled receptors (*Gpr137b*, *Grp171,* and *Gpr183*), and mannose receptor c-type 1 [*Mrc1* (*Cd206*)]. In contrast, *Cd36*, *Met*, *Sgms2*, *Fn1*, *Zeb2,* and *Arg2* levels were higher in the late macrophage population, suggesting M2-like polarization. Therefore, we asked whether these macrophages provide an immunosuppressive microenvironment during *E. faecalis* infection. Cellular interactome analysis revealed enrichment of the ANNEXIN signaling pathway, particularly Anxa1:Fpr1 and Anxa1:Fpr2 ligand-receptor pairs between M2-like macrophage and endothelial or neutrophil cells (*Figure 4—figure supplement 1I–K*), which inhibits pro-inflammatory cytokine production (*Yang et al., 2009*). Together, these results demonstrate that *E. faecalis* infection influences macrophage polarization toward an anti-inflammatory

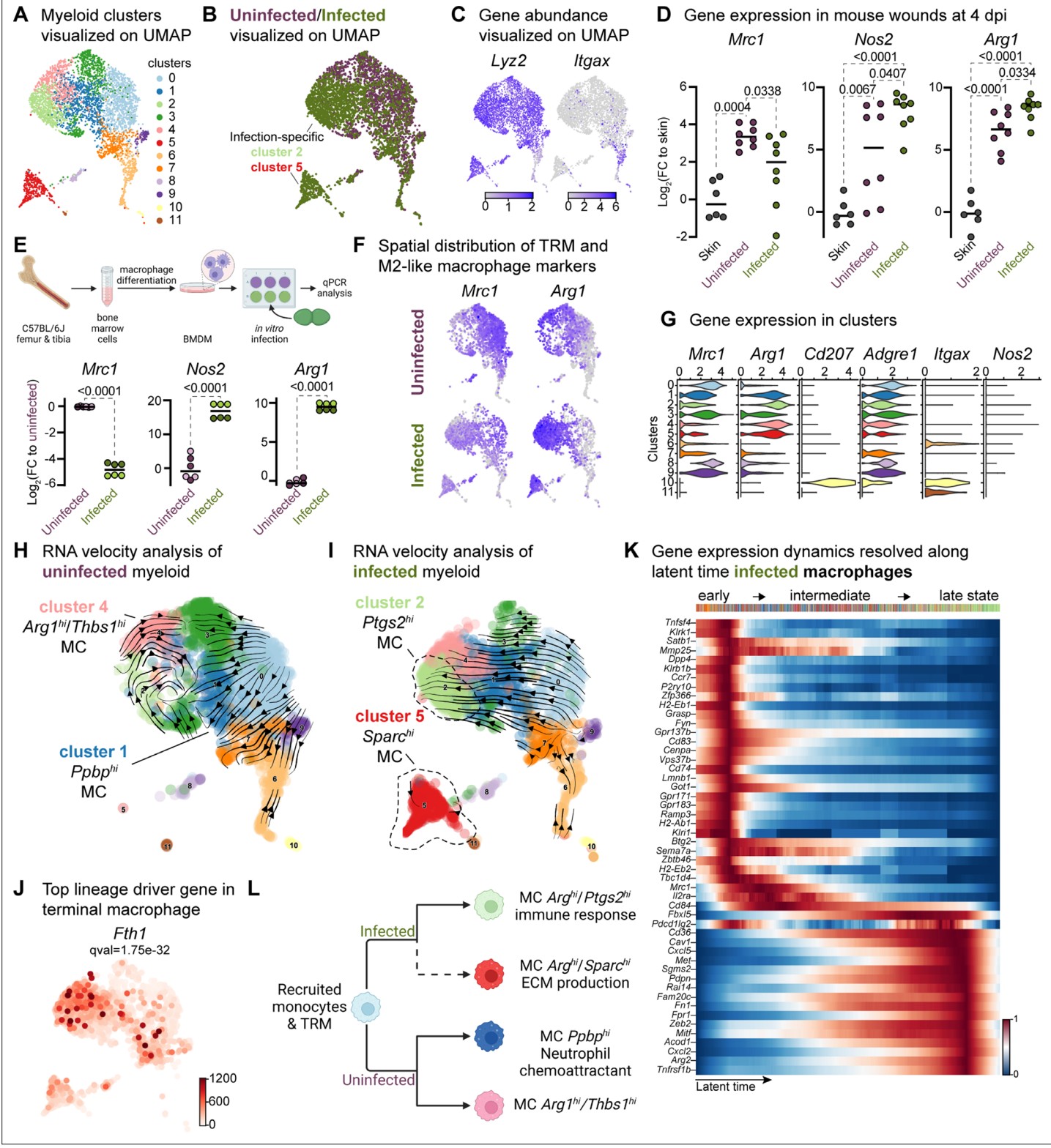

**Figure 4.** Macrophages display M2-like polarization. (**A**) UMAP of the integrated myeloid population reveals nine macrophages, two dendritic cells, and one Langerhans cell cluster. (**B**) Infected myeloid (green) population show unique and shared clusters with the uninfected myeloid (purple) population. (**C**) Spatial distribution of *Lyz2* (macrophage) and *Itgax* (DC and LC) abundance in fibroblasts. (**D**) Gene expression *Mrc1*, *Nos2*, and *Arg1* in mouse wounds at 4dpi, normalized to homeostatic skin (n=6 for skin; n=8 for wounds; one-way ANOVA). (**E**) In vitro infection of unpolarized (**M0**) bone-marrow-derived murine macrophages (BMDMs) resulted in a down-regulation of TRM-associated marker *Mrc1* and an upregulation of M2-like markers *Nos2*

*Figure 4 continued on next page*

*Figure 4 continued*

and *Arg1*. Data are pooled from two biological replicates (shown in light and dark circles, respectively) of three technical replicates each. (**F**) Spatial distribution of TRM and M2-like macrophage markers, *Mrc1* and *Arg1*, respectively. (**G**) Expression of *Mrc1*, *Adgre1*, *Arg1*, *Itgax*, *Cd207*, and *Nos2* in the integrated myeloid dataset. (**H–I**) Dynamic RNA velocity estimation of uninfected (**H**) and infected (**I**) myeloid cells. (**J**) The top lineage driver gene, *Fth1*, was ubiquitously expressed in terminal macrophage populations. (**K**) Putative driver genes of infected macrophages. (**L**) The proposed model describes macrophage characteristics, where neutrophil-attracting and wound repair-associated macrophages were involved in uninfected wound healing. In contrast, bacteria-infected wounds are enriched in efferocytotic macrophages and matrix-producing macrophages. Panels E and L created with BioRender.com, and published using a CC BY-NC-ND license with permission.

The online version of this article includes the following figure supplement(s) for figure 4:

**Figure supplement 1.** The mega myeloid cell population displays M2-like polarization signatures.

phenotype. Importantly, the ANXA1:FPR1 interaction has been implicated in the tumor microenvironment of several malignancies (*Cheng et al., 2014*; *Takaoka et al., 2018*; *Vecchi et al., 2018*; *Zhao et al., 2022*), indicating that the *E. faecalis*-infected wound niche mimics the anti-inflammatory transcriptional program of the tumor microenvironment.

## Neutrophils contribute to the anti-inflammatory microenvironment

Our findings revealed an exacerbated inflammatory phenotype during infection (*Figures 2F and G, 3K and 4J*), specifically marked by an abundance of neutrophils (*Chong et al., 2017*; *Figure 1D*). To facilitate a comparison of the broader immune niche, we focused on the neutrophil population, identifying six subpopulations in the integrated dataset (*Figure 5A* and *Supplementary file 9*). Among these, clusters 0 and 2 were abundant in the infected condition (*Figure 5B* and *Figure 5—figure supplement 1A*). The expression of granulocyte colony-stimulating factor receptor (*Csf3r*) and integrin subunit alpha M [*Itgam* (*Cd11b*)] was highly expressed across all neutrophils (*Figure 5C*). Furthermore, the higher expression of chemokine (CXC motif) receptor 2 (*Cxcr2*), Fc gamma receptor III [*Fcgr3* (*Cd16*)], and ferritin heavy chain (*Fth1*) in clusters 0, 1, 2, 4, and 5 indicated the presence of migrating and maturing neutrophils, regardless of infection status (*Figure 5D*). By contrast, cluster 3 did not express *Cxcr2* and *Fcgr3* had lower expression of *Fth1* but was enriched in calcium/calmodulin-dependent protein kinase type ID (*Camk1d*), suggesting that this neutrophil subpopulation recruited to the infected wound site might be mature but inactive (*Chen et al., 2023*; *Evrard et al., 2018*).

The infection-enriched neutrophils demonstrated a functional shift. Cluster 0 with higher expression of *Lrg1*, indicates an activated phagocytic phenotype (*Figure 5E–G*). By contrast, *Csf1*$^{hi}$ neutrophils (cluster 2) were associated with the negative regulation of apoptosis, cytokine production and neutrophil activation. *Csf1* upregulation in neutrophils mediates macrophage differentiation toward an immunotolerant phenotype (*Braza et al., 2018*), characterized by low expression of the lymphocyte antigen complex (LY6C) and increased proliferation rate. We then inspected the infected myeloid cells and observed a higher abundance of the cell proliferation marker, *Mki67* (Ki-67), and the expression of lymphocyte antigen 6 complex locus 1 (*Ly6c1*). The expression of *Mki67* was specific to terminal macrophage population clusters 2 and 5 (*Figure 4—figure supplement 1G*) and the level of *Ly6c1* was lower in all macrophages but was specific to LCs (*Figure 4—figure supplement 1H*). These findings suggest that neutrophils may contribute to the anti-inflammatory polarization of *E. faecalis* infected macrophages.

RNA velocity analysis also estimated infected neutrophils with a propensity to become *Lrg1*$^{hi}$ neutrophils (*Figure 5H*, cluster 0), driven by genes *Entpd1* (*Cd39*), *Picalm*, *Hdc*, *Cytip*, *Sipa1l1*, and *Fos* (*Figure 5I*, *Figure 5—figure supplement 1B and C*). Moreover, suppression of *Nfkbia* and chemokine ligand-2 (*Cxcl2*) and upregulation of calprotectin (*S100a9*) together indicate an inflammatory response to *E. faecalis* infection (*Figure 5—figure supplement 1B*). To identify molecular changes that could drive neutrophil state transitions, we sorted the genes based on their peak in pseudo-time, revealing the three distinct neutrophil states: early-stage, intermediate-stage, and late-stage, in the *E. faecalis*-infected wounds (*Figure 5J*). The early response was characterized by *Ccl6*, *Il36g* (*Il1f9*), *Cxcl2*, *S100a6*, and *S100a9*, suggesting an initial pro-inflammatory neutrophil response. By contrast, the late stage demonstrated higher expression of ribosomal proteins (*Rpl5*, *Rpl19*, and *Rpl26*), migratory metalloproteinases (*Mmp3*, *Mmp19*, and *Adamts5*), and CXC motif ligand-1 (*Cxcl1*), suggesting perturbed neutrophil infiltration. MMP19 has also been implicated in M2 polarization (*Fingleton*,

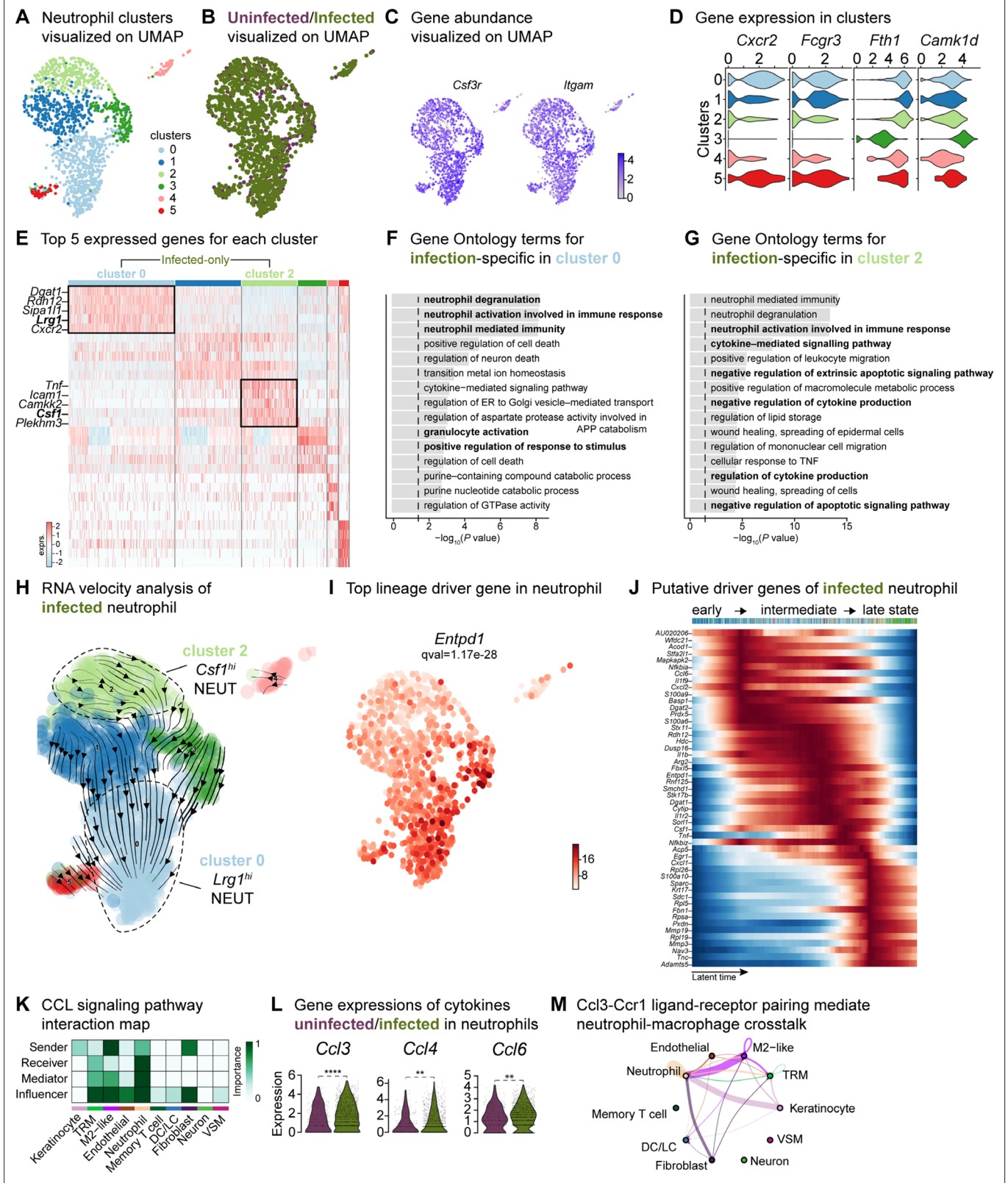

**Figure 5.** Crosstalk between neutrophils and anti-inflammatory macrophages regulates the CCL signaling pathway. (**A**) UMAP of the integrated myeloid population reveals six Louvain clusters. (**B**) The infected neutrophil (green) population shows unique and shared clusters with the uninfected neutrophil (purple) population. (**C**) Spatial organization of *Csf3r* and *Itgam* abundance in neutrophils. (**D**) Expression of *Cxcr2*, *Fcgr3*, *Fth1*, and *Camk1d* in Louvain clusters. (**E**) Heat map of top 5 differentially expressed marker genes in neutrophils. Rectangle boxes indicate infection-specific Louvain clusters. (**F–G**)

*Figure 5 continued on next page*

*Figure 5 continued*

The bar plots show the top 15 Gene Ontology terms for infection-specific (**F**) *Lrg1*hi (cluster 0) and (**G**) *Csf1*hi (cluster 2) populations. (**H**) Dynamic RNA velocity estimation of infected neutrophils. (**I**) The top lineage driver gene, *Entpd1*, was ubiquitously expressed in *Lrg1*hi neutrophils (cluster 0). (**J**) Putative driver genes of infected neutrophil clusters. (**K**) Cytokine signaling pathway (CCL) cell-cell interaction map. (**L**) Gene expression of cytokines *Ccl3*, *Ccl4*, and *Ccl6* in neutrophils. (**M**) Ccl3:Ccr1 ligand-receptor interaction mediate neutrophil-macrophage crosstalk.

The online version of this article includes the following figure supplement(s) for figure 5:

**Figure supplement 1.** Neutrophil infiltration was enhanced in response to *E. faecalis* infection.

*2017*), consistent with the anti-inflammatory signatures of the infected macrophage populations (*Figure 4J* and *Figure 4—figure supplement 1E*). Cellular interactome analysis further predicted the anti-inflammatory status of neutrophils by a strong correlation in the CCL signaling pathway between neutrophils and M2-like macrophages, particularly through the Ccl3:Ccr1 axis (*Figure 5K–M* and *Figure 5—figure supplement 1D*), an interaction predictive of neutrophil extravasation during *E. faecalis* infection (*Hautz et al., 2023*). Neutrophil extravasation is a vital event in immune responses to infections to ensure the survival of the host (*Theocharidis et al., 2022*). In summary, our single-cell analysis of neutrophils during *E. faecalis* wound infection revealed a perturbed pro-inflammatory resolution during infection that may contribute to anti-inflammatory macrophage polarization. Furthermore, our findings uncover prominent differences in immune cell composition in infected wounds, featuring *Lrg1*-rich neutrophil abundance (cluster 0), together with the enrichment of *Arg1*hi M2-like macrophage polarization (clusters 2 and 5). As such, wound healing during *E. faecalis* is characterized by a dysregulated immune response compared to uninfected wounds, which could be associated with delayed healing or chronic infection.

## Anti-inflammatory macrophages induce pathogenic angiogenesis

Based on the role of angiogenesis in tissue repair, and our observations of the anti-inflammatory signatures provided by keratinocytes, fibroblasts, and macrophages, we investigated their impact on endothelial cells (ECs). First, we analyzed the two endothelial cell (EC) populations in the integrated dataset and identified 13 clusters with high *Pecam1* and *Plvap* expression. Notably, clusters 0 and 8 were exclusively found in the infected ECs. (*Figure 6A and B*, and *Figure 6—figure supplement 1A–C*). These clusters were involved in ECM deposition, cell differentiation, and development, indicating an anti-inflammatory niche (*Figure 6C and D*). Interestingly, RNA velocity analysis of infected ECs showed a faster velocity in the *Sparc*hi (cluster 0) and *Cilp*hi (cluster 8) cells, suggesting a dynamic transcriptional state (*Figure 6E*). The top lineage driver genes *Malat1*, *Tcf4*, *Rlcb1*, *Diaph2*, *Bmpr2*, and *Adamts9* indicate a pathogenic mechanism in proliferating ECs (*Figure 6F*).

We then explored whether the anti-inflammatory characteristics observed in infected epithelial cells, fibroblasts, and immune cells impacted ECs. To predict these cellular interactions, we conducted a differential NicheNet analysis (*Browaeys et al., 2020*), which involved linking the expression of ligands with corresponding receptors and downstream target gene expressions of these pairs in our scRNA-seq atlas of infected cells. Remarkably, NicheNet analysis revealed Il1b ligand of M2-like macrophages, correlated with the expression of target genes such as biglycan (*Bgn*), *Cd14*, *Ccl3*, *Csf3r*, *Itgam*, and *Tnf* in ECs (*Figure 6G* and *Figure 6—figure supplement 1D*), in line with previous studies (*Perrault et al., 2018*; *Vu et al., 2022*). Notably, BGN, a proteoglycan, has been associated with tumor EC signatures (*Cong et al., 2021*; *Morimoto et al., 2021*), further supporting the anti-inflammatory role of M2-like macrophages observed in the infected dataset. Moreover, the infection-induced expression of *Cd14* in EC suggests Toll-like receptor (TLR) activation (*Dauphinee and Karsan, 2006*; *Lloyd and Kubes, 2006*). Cell-cell interaction analysis also predicted that the Ptgs2, Spp1 and Il1a ligand activity in M2-like macrophages influenced the expression of target genes such as calprotectin (*S100a8* and *S100a9*) and colony-stimulating factor 3 (*Csf3*). These interactions suggest a potential disruption of EC integrity in the presence of *E. faecalis* infection. Similarly, according to the CellChat analysis, ECs exhibited activation of the SPP1 and CXCL signaling pathways (*Figure 6H–J*, *Figure 6—figure supplement 1E and F*). Expression of the ligands Spp1, Cxcl2, and Cxcl3 were abundant in keratinocytes, fibroblasts, and M2-like macrophages during the infection (*Figure 6J*). In the uninfected ECs, however, fibroblasts modulated the expression of target genes such as TEK tyrosine kinase (*Tek*), Forkhead box protein O1 (*Foxo1*) and selectin-E (*Sele*), with notable angiopoietin 1 activity (*Figure 6—figure supplement 1D*), pointing to normal endothelial cell activity.

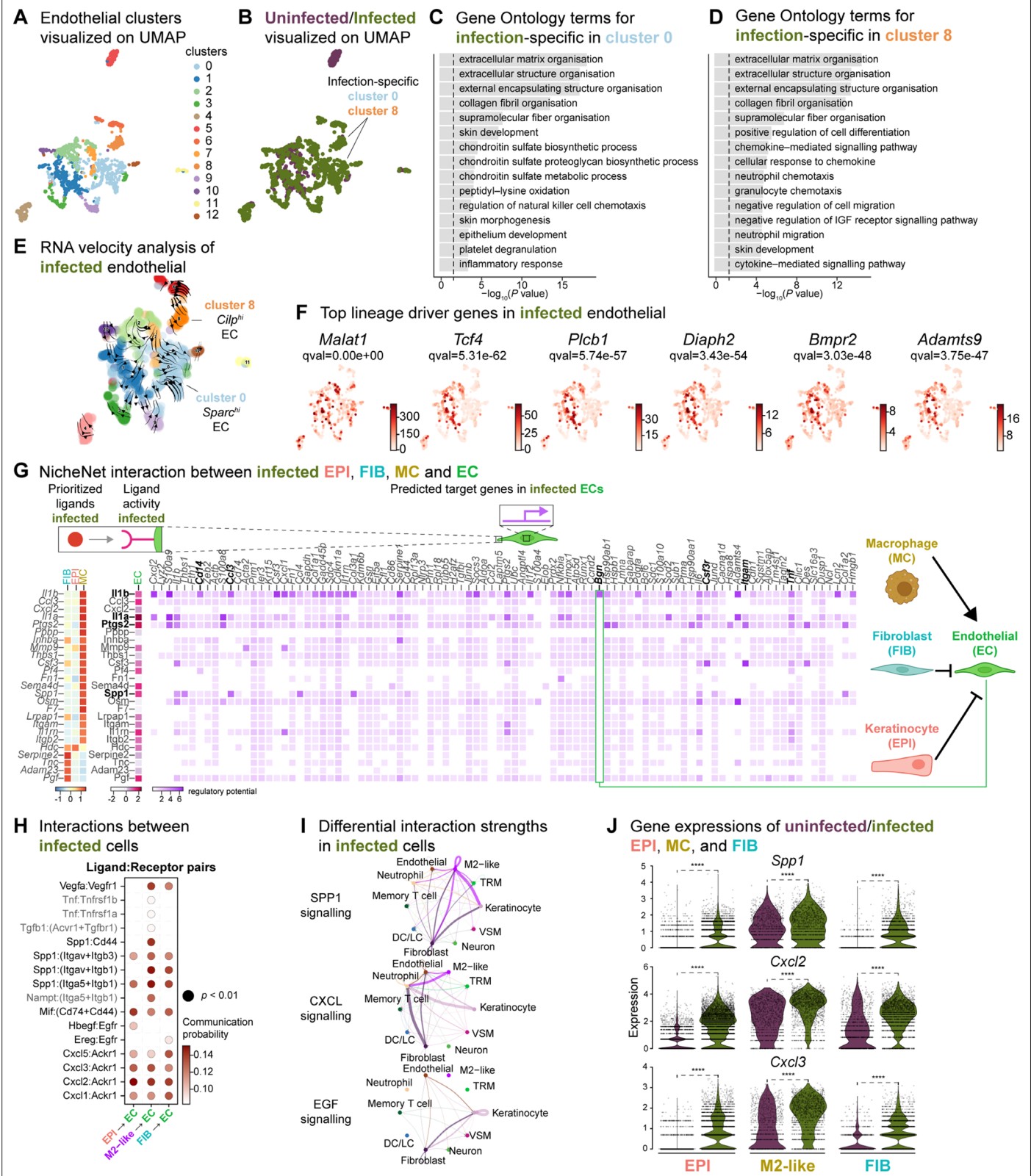

**Figure 6.** Macrophage-EC interactions display an anti-inflammatory niche. (**A**) UMAP of integrated endothelial cells reveals 13 clusters. (**B**) Infected endothelial cells (green) show unique and shared clusters with uninfected endothelial cells (purple). (**C–D**) Gene Ontology analysis of infection-specific clusters 0 (**C**) and 8 (**D**). (**E**) Dynamic RNA velocity estimation of infected endothelial cells. (**F**) The top lineage driver genes, *Malat1*, *Tcf4, Plcb1, Diaph2, Bmpr2,* and *Adamts9* in infected endothelial cells. (**G**) NicheNet interaction heat map between keratinocytes, fibroblasts, macrophages, and endothelial

*Figure 6 continued on next page*

*Figure 6 continued*

cells. Note the macrophage *Il1b*-specific *Bgn* induction in infected ECs. (**H**) The dot plot depicts interactions between endothelial cells (receptors) and keratinocytes, fibroblasts, and macrophages (ligands). Rows demonstrate a ligand-receptor pair for the indicated cell-cell interactions (column). (**I**) Differential interaction strengths of a cellular interactome for TNF, SPP1, and CXCL signaling pathways between all cell types in infection. (**J**) Spp1, Cxcl2, and Cxcl3 gene expression in keratinocytes, M2-like macrophages, and fibroblasts (Wilcoxon Rank Sum test, ****p<0.0001).

The online version of this article includes the following figure supplement(s) for figure 6:

**Figure supplement 1.** Fibroblasts orchestrate a reparative niche in the uninfected wounds.

During unperturbed wound healing, macrophages modulate angiogenesis by producing proteases, including matrix metalloproteinases (MMPs), which help degrade the extracellular matrix in the wound bed. Additionally, macrophages secrete chemotactic factors such as TNF-α, VEGF and TGF-β to promote the EC migration (*Du Cheyne et al., 2020*; *Wilkinson and Hardman, 2020*). Furthermore, the TGF-β signaling pathway induces fibroblast proliferation and ECM production in wound healing (*Cutroneo, 2007*; *Pakyari et al., 2013*). Our analysis of the cellular interactome in the uninfected wound dataset revealed strong interactions in the TGF-β and VEGF signaling pathways (*Figure 6—figure supplement 1H–I*), corroborating previous studies (*Joost et al., 2016*; *Vu et al., 2022*). These findings suggest that uninfected wounds undergo reparative angiogenesis while *E. faecalis* infection evokes pathological vascularization. Overall, our analysis underlines the M2-like macrophage-EC interactions as targets of altered cell-cell signaling during bacteria-infected wound healing.

## Discussion

Wound healing is an intricate process that involves the cooperation of various cellular and extracellular components. Disruptions to this network can perturb the healing dynamics. Previous scRNA-seq studies have primarily focused on the transcriptional profiles of epithelial and fibroblast populations in wound healing (*Deng et al., 2021*; *Haensel et al., 2020*; *Joost et al., 2020*; *Joost et al., 2018*; *Joost et al., 2016*). However, the impact of the host-pathogen interactions on wound healing transcriptional programs remains to be investigated. Here, our work presents the first comprehensive single-cell atlas of mouse skin wounds following infection, highlighting the transcriptional aberration in wound healing. *E. faecalis* has immunosuppressive activity in various tissues and infection sites (*Chong et al., 2017*; *Kao et al., 2023*; *Kao and Kline, 2019*; *Tien et al., 2017*). By surveying approximately 23,000 cells, we identified cell types, characterized shared and distinct transcriptional programs, and delineated terminal phenotypes and signaling pathways in homeostatic healing or bacterial wound infections. These scRNA-seq findings elucidate the immunosuppressive role of *E. faecalis* during wound infections, providing valuable insights into the underlying mechanisms.

To explore the cellular landscape of *E. faecalis*-infected wounds and the impact of infection on wound healing, we focused on cell types specifically enriched in these tissues. Our analysis revealed infection-activated transcriptional states in keratinocytes, fibroblasts, and immune cells. The presence of clusters 0 and 6, exclusive to the epithelial dataset, along with RNA velocity analysis pointing to *Zeb2*-expressing cells (*Figure 2I*), suggests a potential role of *E. faecalis* in promoting a partial EMT in keratinocytes. We also identified a terminal keratinocyte population (cluster 2) at 4 dpi, characterized by the consumption of *Krt77* and *Krtdap* over pseudo-time, confirming terminal keratinocyte differentiation (*Figure 2I* and *Figure 2—figure supplement 1C*). By contrast, uninfected wounds showed proliferating keratinocyte (*Mki67*$^{hi}$) and hair follicle (*Lrg5*$^{hi}$) cells (*Figure 2H*), consistent with previous findings on wound healing and skin maintenance signatures (*Joost et al., 2020*; *Joost et al., 2018*). Furthermore, we found two infection-specific fibroblast populations enriched in *Lyz2* (and *Tagln*) and *Timp1*, identifying their myofibroblast characteristics. The RNA velocity analysis confirmed cluster 5 as the terminal fibroblast population derived from myofibroblasts (cluster 2) (*Figure 3I*). Additionally, an increase in *Timp1* expression correlated with cell growth and proliferation signatures (*Sparc*, *Fgf7* and *Malat1*) over latent time in cluster 5 (*Figure 3—figure supplement 1C and D*). These findings collectively demonstrate a profibrotic state in fibroblasts driven by *E. faecalis*-induced transcriptional dynamics.

To investigate the impact of immunosuppressive signatures in keratinocytes and fibroblasts, we examined how *E. faecalis* infection influenced the immune response in wounds. Given that prolonged macrophage survival is associated with impaired wound healing (*Kim and Nair, 2019*; *Krzyszczyk*

*et al., 2018*), we primarily focused on the macrophage populations. We identified subpopulations of M2-like macrophages expressing *Arg1*, *Ptgs2,* and *Sparc* genes (*Figure 4F–I*). COX-2 (cyclooxygenase-2, encoded by *Ptgs2* gene) has been reported as a crucial regulator of M2-like polarization in tumor-associated macrophages (*Na et al., 2013*; *Wang et al., 2021*). While macrophage polarization is complex in humans and other higher organisms (*Orecchioni et al., 2019*; *Watanabe et al., 2019*), our scRNA-seq data suggests that *E. faecalis* infection drives an anti-inflammatory microenvironment resembling the tumor microenvironment. These findings confirm and expand on previous studies highlighting the immunosuppressive role of *E. faecalis* in different contexts (*Chong et al., 2017*; *Kao and Kline, 2019*; *Tien et al., 2017*).

Our findings also elaborate on the potential cell-cell communication and signaling pathways involved in wound healing. Ligand-receptor interaction analysis revealed an immune-suppressive ecosystem driven by M2-like macrophages during bacterial infection (*Figure 6G*). CellChat analysis confirmed that M2-like macrophages play a crucial role in regulating angiogenesis through the Vegfa:Vegfr1 ligand-receptor pair (*Figure 6H and I*, *Figure 6—figure supplement 1E and F*, and *Supplementary file 6*). Further, the Spp1 ligand correlated with integrins (Itgav+Itgb1, Itgav+Itgb3, Itga5+Itgb1) and the cluster of differentiation receptor Cd44 in ECs. Notably, while the Spp1 ligand of macrophages registered strong interactions with ECs, infected keratinocytes and fibroblasts were identified as the primary sources of Spp1 ligand. The infection-specific abundance of *Spp1* in these clusters highlight a partial EMT state in our extended analyses (*Figure 2E–I*). Furthermore, the interaction between Spp1 and Cd44 in M2-like macrophages and ECs indicated a proliferative state. The neutrophil chemokine *Cxcl2*, which promotes wound healing and angiogenesis (*Girbl et al., 2018*; *Sawant et al., 2021*), strongly correlated with the atypical chemokine receptor *Ackr1*, suggesting neutrophil extravasation. Additionally, the secretion of heparin-binding epidermal growth factor (*Hbegf*) by keratinocytes and epiregulin-enriched fibroblasts (*Ereg^hi*) further confirmed the presence of an anti-inflammatory microenvironment and angiogenic ECs in bacteria-infected wounds (*Figure 6H and I*, and *Figure 6—figure supplement 1F*).

In summary, our study provides insights into the cellular landscape, transcriptional programs, and signaling pathways associated with uninfected and *E. faecalis*-infected skin wounds. We confirm the immune-suppressive role of *E. faecalis* in wound healing, consistent with previous findings in different experimental settings (*Chong et al., 2017*; *Kao et al., 2023*; *Tien et al., 2017*). Importantly, we identify specific ligand-receptor pairs and signaling pathways affected during wound infection. The increased number of predicted signaling interactions suggests that *E. faecalis* modulates cellular communication to alter the immune response. Notably, the CXCL/SPP1 signaling pathway emerges as a critical player in shaping the immune-altering ecosystem during wound healing, highlighting its potential as a therapeutic target for chronic infections. Collectively, our findings demonstrate the collaborative role of keratinocytes, fibroblasts, and immune cells in immune suppression through CXCL/SPP1 signaling, providing new avenues for the treatment of chronic wound infections.

## Limitations of the study

Our study provides a comprehensive comparison of the transcriptomic and cellular communication profiles between uninfected and *E. faecalis*-infected full-thickness mouse skin wounds at 4 days post-infection. However, our study lacks a reference dataset of uninfected, unwounded dorsal skin transcriptome, which would have allowed us to investigate transcriptomic changes temporally, especially induced by the wounding alone. Although we explored public datasets to address this limitation (*Haensel et al., 2020*; *Vu et al., 2022*), the low number of cells in public datasets, particularly for macrophage populations (data not shown), prevented their inclusion in our analysis. Additionally, the absence of multiple time points hinders our ability to examine temporal changes and the dynamic kinetics of host responses following infection.

# Methods

**Key resources table**

| Reagent type (species) or resource | Designation | Source or reference | Identifiers | Additional information |
|---|---|---|---|---|
| Strain, strain background (*Enterococcus faecalis*) | OG1RF | *Dunny et al., 1978* | *Enterococcus faecalis* OG1RF | |
| Strain, strain background (*Mus musculus*) | Mouse C57BL/6 | InVivos | n/a | |
| Peptide, recombinant protein | Bovine serum albumin | Merck | A7030 | |
| Peptide, recombinant protein | Collagenase type I | Thermo Fisher Scientific | 17100017 | |
| Peptide, recombinant protein | Dispase II | Thermo Fisher Scientific | 17105041 | |
| Peptide, recombinant protein | Liberase TM | Merck | 5401119001 | |
| Peptide, recombinant protein | Recombinant Mouse M-CSF | BioLegend | 576404 | |
| Chemical compound, drug | Isoflurane | Vetpharma Animal Health | NA | |
| Chemical compound, drug | Nair Moisturising Hair Removal Cream | Chruch and Dwight Co | NA | |
| Commercial assay, kit | Chromium Next GEM Chip G Single Cell Kit | 10 x Genomics | 1000127 | |
| Commercial assay, kit | Chromium Next GEM Single Cell 3' Gel Bead Kit v3.1 | 10 x Genomics | 1000129 | |
| Commercial assay, kit | Chromium Next GEM Single Cell 3' GEM Kit v3.1 | 10 x Genomics | 1000130 | |
| Commercial assay, kit | Chromium Next GEM Single Cell 3' Kit v3.1 | 10 x Genomics | 1000269 | |
| Commercial assay, kit | Dual Index Kit TT Set A | 10 x Genomics | 1000215 | |
| Commercial assay, kit | EZ-10 DNAaway RNA Mini-Preps kit | Bio Basic | BS88136-250 | |
| Commercial assay, kit | Library Construction Kit | 10 x Genomics | 1000196 | |
| Commercial assay, kit | Luna SYBR Green | New England Biolabs | M3003 | |
| Commercial assay, kit | RevertAid Reverse Transcriptase | Thermo Fisher | 01327685 | |
| Commercial assay, kit | Tube, Dynabeads MyOne SILANE | 10 x Genomics | 2000048 | |
| Software, algorithm | anndata 0.8.0 | PyPI | https://pypi.org | |
| Software, algorithm | BioRender | https://biorender.com | n/a | |
| Software, algorithm | CellChat 1.6.1 | *Jin et al., 2021* | | |
| Software, algorithm | cellrank 1.5.1 | *Lange et al., 2022* | | |
| Software, algorithm | Cell Ranger 6.1.2 | 10 X Genomics, Inc. | https://www.10xgenomics.com | |
| Software, algorithm | clustree 0.5.0 | *Zappia and Oshlack, 2018* | | |
| Software, algorithm | glmGamPoi 1.8.0 | *Ahlmann-Eltze and Huber, 2021* | | |
| Software, algorithm | miQC 1.4.0 | GitHub; *Greene Laboratory, 2022* | https://github.com/greenelab/miQC | |
| Software, algorithm | NicheNet | *Browaeys et al., 2020* | | |
| Software, algorithm | numpy 1.23.5 | PyPI | https://pypi.org | |
| Software, algorithm | pandas 1.5.3 | PyPI | https://pypi.org | |

*Continued on next page*

*Continued*

| Reagent type (species) or resource | Designation | Source or reference | Identifiers | Additional information |
|---|---|---|---|---|
| Software, algorithm | PanglaoDB | PanglaoDB | https://panglaodb.se/markers.html | |
| Software, algorithm | Prism 9.4.1 | GraphPad Software | https://www.graphpad.com | |
| Software, algorithm | PyCharm 2022.3.3 | JetBrains | https://www.jetbrains.com | |
| Software, algorithm | Python 3.9.16 | Python Software Foundation | https://python.org | |
| Software, algorithm | R 4.2.1 | The R Foundation | https://www.r-project.org | |
| Software, algorithm | RStudio 2022.07.2+576 | Posit Software | https://posit.co | |
| Software, algorithm | samtools 1.13 | GitHub; *Ohan and samtools, 2021* | https://github.com/samtools | |
| Software, algorithm | scanpy 1.9.1 | PyPI | https://pypi.org | |
| Software, algorithm | scipy 1.10.0 | PyPI | https://pypi.org | |
| Software, algorithm | scDblFinder 1.10.0 | *Germain et al., 2021* | | |
| Software, algorithm | sctransform 0.3.5 | *Choudhary and Satija, 2022* | | |
| Software, algorithm | scvelo 0.2.5 | *Bergen et al., 2020* | | |
| Software, algorithm | Seurat 4.3.0 | *Hao et al., 2021* | | |
| Software, algorithm | UMAP | *McInnes et al., 2018*; *McInnes, 2024* | https://github.com/lmcinnes/umap | |
| Software, algorithm | velocyto.py 0.17.17 | *La Manno et al., 2018* | | |
| Software, algorithm | WGCNA 1.72–1 | *Langfelder and Horvath, 2008* | | |

## Resource availability

### Lead contact

Further information and requests for resources should be directed to and will be fulfilled by the lead contacts, Kimberly Kline, at kimberly.kline@unige.ch and Guillaume Thibault, at thibault@ntu.edu.sg.

### Material availability

This study did not generate new reagents.

## Method details

### Mouse wound infection model

In vivo procedures were approved by the Animal Care and Use Committee of the Biological Resource Centre (Nanyang Technological University, Singapore) in accordance with the guidelines of the Agri-Food and Veterinary Authority and the National Advisory Committee for Laboratory Animal Research of Singapore (ARF SBS/NIEA-0314). Male C57BL/6J mice were housed at the Research Support Building animal facility under pathogen-free conditions. Mice between 5 and 7 weeks old (22–25 g; InVivos, Singapore) were used for the wound infection model, modified from a previous study (*Keogh et al., 2016*). Briefly, mice were anesthetized with 3% isoflurane. Dorsal hair was removed by shaving, followed by hair removal cream (Nair cream, Church and Dwight Co) application. The shaven skin was then disinfected with 70% ethanol and a 6 mm full-thickness wound was created using a biopsy punch (Integra Miltex, New York). Two mice were infected with 10 μl of $2 \times 10^8$ bacteria/ml inoculum (*Enterococcus faecalis* OG1RF strain), while the other two served as controls for uninfected wounds. All wounds were sealed with transparent dressing (Tegaderm, 3M, St Paul Minnesota). Mice were euthanized 4 days post-infection (4 dpi), and wounds collected immediately using a biopsy punch

with minimal adjacent healthy skin were stored in Hanks' Buffered Salt Solution (HBSS; $Ca^{2+}/Mg^{2+}$-free; Sigma Aldrich, Cat. #H4385) supplemented with 0.5% bovine serum albumin (BSA; Sigma Aldrich, Cat. #A7030).

## In vitro culture and infection of bone-marrow-derived macrophages

Murine bone-marrow-derived macrophages (BMDMs) were isolated from bone marrow cells of 7–8 weeks old C57BL/6J male mice as described previously (*Toda et al., 2021*), except that BMDMs were differentiated in 100 mm non-treated square dishes (Thermo Scientific, Singapore) with supplementation of 50 ng/ml M-CSF (BioLegend, Singapore). On day 6, differentiated BMDMs were harvested by gentle cell scraping in Dulbecco's PBS (DPBS; Gibco, Singapore), and $10^6$ BMDMs were seeded in six-well plates using bone marrow growth medium without antibiotics. Following overnight incubation, the growth medium was replaced with complete DMEM (DMEM supplemented with 10% FBS) for *E. faecalis* infection. Log-phase *E. faecalis* OG1RF cultures were washed and normalized to an $OD_{600}$/ml of 0.5 in complete DMEM, equivalent to approximately $3 \times 10^8$ CFU/ml. BMDMs were then infected with OG1RF at a multiplicity of infection (MOI) of 10 for 1 hr, followed by centrifugation at 1000 x *g* for 5 min prior to incubation to promote BMDM-bacteria contact. For uninfected controls, complete DMEM was added instead of MOI 10 bacterial suspension. At 1 hpi, BMDM were washed thrice with DPBS and incubated in complete DMEM supplemented with 10 μg/ml vancomycin and 150 μg/ml gentamicin for 23 hr to kill extracellular bacteria, until a final time point of 24 hpi.

## Quantitative qPCR analysis

Total mRNA was extracted using the TRIzol reagent (Invitrogen, cat. # 15596018) and purified using an EZ-10 DNAaway RNA Mini-Preps Kit (Bio Basic, cat. #BS88136-250). Complementary DNA (cDNA) was synthesized from total RNA using RevertAid Reverse Transcriptase (Thermo Fisher, Waltham, MA, cat. # 01327685) according to the manufacturer's protocol. Real-time PCR was performed using Luna SYBR Green (New England Biolabs, UK) according to the manufacturer's protocol using a CFX-96 Real-time PCR system (Bio-Rad, Hercules, CA, USA). A 100 ng of cDNA and 0.25 μM of the paired primer mix for target genes were used for each reaction. Relative mRNA was normalized to the housekeeping gene glyceraldehyde-3-phosphate dehydrogenase (*Gapdh*) or a geometric mean of Ct values from beta-actin (*Actb*) and ubiquitin (*Ubc*) housekeeping genes, as calculated by BestKeeper (*Pfaffl et al., 2004*) and log2 fold changes were plotted with reference to the uninfected, unwounded skin. The primer pairs used in this study are listed as follow:

| Gene | Forward | Reverse |
| --- | --- | --- |
| *ActB* | ATC AGC AAG CAG GAG TAC GAT | GTG TAA AAC GCA GCT CAG TAA CA |
| *Arg1* | CAG AAG AAT GGA AGA GTC AG | CAG ATA TGC AGG GAG TCA CC |
| *Egf* | TGG CTC GAA GTC AGA TCC ACA | TTC TCG GGC ACA TGG TTA ATG |
| *Gapdh* | TCA GGA GAG TGT TTC CTC GTC CC | TCT CGG CCT TGA CTG TGC CG |
| *Fgf1* | CCC TGA CCG AGA GGT TCA AC | GTC CCT TGT CCC ATC CAC G |
| *Mrc1* | TTC AGC TAT TGG ACG CGA GG | GAA TCT GAC ACC CAG CGG AA |
| *Nos2* | TGT CGC AGC TCC CTA TCT TG | GGA AGC CAC TGA CAC TTC GC |
| *Pdgfa* | TGG CTC GAA GTC AGA TCC ACA | TTC TCG GGC ACA TGG TTA ATG |
| *Tgfb1* | GGA AAT CAA CGG GAT CAG CCC | GCT GCC GCA CAC AGC AGT TC |
| *Ubc* | CCC AGT GTT ACC ACC AAG AAG | CCC CAT CAC ACC CAA GAA CA |

## Dissociation of tissue and library preparation

Tissues were transferred to a sterile 10 $mm^2$ tissue culture dish and cut into small fragments using a sterile blade. The dissociation cocktail [10 mg/ml of dispase II (GIBCO, Cat. #17105041), 250 mg/ml of collagenase type I (GIBCO, cat. #17100017), and 2.5 mg/ml of liberase TM (Roche, cat. #5401119001)] was dissolved in HBSS. The tissue was digested enzymatically in pre-warmed dissociation buffer for 2 hr at 37 °C with manual orbital shaking every 15 min and then the dissociated cells were sifted

through a sterile 70 μm cell strainer (Corning, cat. #352350) on ice and washed thrice with HBSS supplemented with 0.04% BSA. The remaining undigested tissue on the filter was re-incubated in 0.05%[w/v] trypsin/EDTA (GIBCO, cat. #25–051 CI) for 15 min at 37 °C. Trypsin-digested cells were pooled with cells on ice by sifting through a sterile 70 μm cell strainer. The pooled cell suspension was washed thrice with HBSS supplemented with 0.04% BSA, followed by final filtering using a sterile 40 μm cell strainer (Corning, cat. #352340). The cell suspensions were centrifuged at 300 x *g* for 10 min, and the pellets were resuspended in Dulbecco's phosphate-buffered saline (DPBS; Thermo Fisher Scientific, cat. #14190094). Cell viability was determined by using Countess 3 FL Automated Cell Counter (Invitrogen) by mixing cell suspension with 0.4% trypan blue stain (Invitrogen, cat. #T10282) at a ratio of 1:1 for a minimum of four counts per sample. Isolated cells were encapsulated for single-cell RNA sequencing with microfluidic partitioning using the Chromium Single Cell 3' Reagent Kits (v3.1 Chemistry Dual Index, Protocol #CG000315), targeting 8000 cells (10 X Genomics, Pleasanton, CA). Libraries then were pooled by condition and sequenced on the HiSeq6000 system by NovogeneAIT (Singapore). A detailed protocol can be found at protocols.io (10.17504/protocols. io.yxmvmn8m9g3p/v1).

## Data processing

Raw data (.bcl) was used as an input to *Cell Ranger* (v6.1.2, 10X Genomics) with *mkfastq* function for trimming, base calling, and demultiplexing of reads by NovogeneAIT (Singapore). FASTQ files were aligned, filtered, barcoded and UMI counted using `cellranger count` command using *GENCODE vM23/Ensembl 98* (https://cf.10xgenomics.com/supp/cell-exp/refdata-gex-GRCh38-2020-A.tar.gz) mouse reference genome with `--expect-cells` of 8000 per sample on the National Supercomputer Centre Singapore (NSCC) High Power Computing platform (ASPIRE 1). The raw data can be found at Gene Expression Omnibus with the accession number GSE229257.

## Integration and downstream analysis

Datasets from each sample were integrated using *Seurat* version 4.3.0 (*Hao et al., 2021*) on *R* (v4.2.1) using *RStudio* interface (RStudio 2022.07.2+576 "Spotted Wakerobin" release) on macOS Ventura (13.0.1). Briefly, Seurat objects were created based on the criteria of a minimum of three cells expressing a minimum of 200 features. Low-quality cells were determined by a probabilistic approach using the *miQC* package (v1.4.0). Similarly, doublets were removed with *scDblFinder* v1.10.0 (*Germain et al., 2021*). Cell cycle stages were determined by using *Seurat*'s built-in updated cell-cycle gene list. Feature counts were normalized using `SCTransform()` v2 regularization (*Choudhary and Satija, 2022*) with a scaling factor of 1x10⁴ and Gamma-Poisson Generalized Linear Model by `glmGamPoi()` function (*Ahlmann-Eltze and Huber, 2021*), during which cell cycles and mitochondrial genes were regressed out for integration of 3,000 anchors. Fourteen principal components (PC) were included in dimension reduction calculated by the point where the change of percentage of variation was more than 0.1% between the two consecutive PCs. First, *k*-Nearest Neighbors were computed using *pca* reduction, followed by identifying original Louvain clusters with a resolution of 0.7 calculated by *clustree* v0.5.0 (*Zappia and Oshlack, 2018*). Uniform Manifold Approximation and Projection (UMAP) was used for visualization throughout the study (*McInnes et al., 2018*). A log 2-fold-change of 0.5 was set to identify gene expression markers for all clusters with `min.pct` of 0.25. Density plots were generated in the *Nebulosa* (v1.6.0) package to demonstrate specific cell population signatures. Finally, we computed weighted gene co-expression networks for describing the correlation patterns among genes across samples (*Langfelder and Horvath, 2008*). We also compared the 2D UMAP representations of the two conditions. Using the integrated dataset, we split the dataset based on the conditions by the built-in `SplitObject()` function of the Seurat package. Then, we computed a cross-entropy test for the UMAP projections by applying a two-sided Kolmogorov-Smirnov test (*Roca et al., 2023*).

## Cell type annotation

An unbiased cell annotation was conducted at two levels using the *ScType* algorithm with slight modifications (*Ianevski et al., 2022*). First, we identified mega classes using gene set signatures on PanglaoDB (https://panglaodb.se/; Acc. date August 2022). Secondly, we further detailed each cluster based on published comprehensive mouse skin datasets (*Joost et al., 2020*; *Joost et al., 2016*). The processed Seurat object can be found at Zenodo (https://doi.org/10.5281/zenodo.7608212).

## Cell-cell interactions

*NicheNet*'s differential *R* implementation (v1.1.1) was used to interrogate cell-cell interactions (*Browaeys et al., 2020*). Three databases provided, namely *ligand-target prior model*, *ligand-receptor network*, and *weighted integrated networks*, were used *NicheNet*'s *Seurat* wrapper where endothelial cells were set as the receiver (receptor) cell population, and keratinocytes, fibroblasts, and macrophages were defined as the sender (ligand) populations. The *NicheNet* heat map was plotted to visualize the ligand activity-target cell gene expression matrix. We further compared cell-cell interactions between different niches to better predict niche-specific ligand-receptor (L-R) pairs using differential *NicheNet* analysis. *CellChat* (v1.6.1) was used to further identify secreted ligand-receptor pairs across all cell populations (*Jin et al., 2021*).

## RNA velocity analysis

BAM files generated during the alignment of the raw data in Cell Ranger, were sorted by cell barcodes with `sort -t CB` using *samtools* (v1.13) in the command line. Spliced and unspliced reads were annotated in *velocyto.py* (v0.17.17) pipeline (*La Manno et al., 2018*), with repeats mask annotation file downloaded from https://genome.ucsc.edu/ for mouse reference genome to generate `.loom` files. To solve full transcriptomics dynamics, we used the generalized dynamic model on *scVelo* v0.2.0 (*Bergen et al., 2020*) in *Python* 3 (v3.9.16) virtual environment set up in PyCharm (Education License v2022.3.3, build PY-223.8836.43; JetBrains, Czechia), and visualized main cell populations and subset analysis with velocity stream plots and individual genes that drive RNA velocity. Lineage driving genes were computed in *CellRank* v1.5.1 (*Lange et al., 2022*). Annotated data (`anndata`) files for each cell population and condition can be found at https://doi.org/10.5281/zenodo.7608772. The R scripts and complete *Jupyter* notebooks are also available at https://github.com/cenk-celik/Celik_et_al, (copy archived at *Celik et al., 2024*).

## Statistical analysis

Statistical analyses were performed in *R*, python and Prism 9 (v9.4.1, GraphPad) whichever was applicable for the type of analysis. p Values were adjusted with the Benjamini-Hochberg method for high-dimensional data analysis. The details of the significance and type of the test were indicated in the figure legends. Data are expressed as median ± SEM. A p-value of 0.05 was set as a significance threshold unless stated otherwise.

## Acknowledgements

We are grateful to Drs. Claudia Stocks and Sophie Janssens for helpful discussions and critical reading of the manuscript. We thank the Singapore Centre for Environmental Life Sciences Engineering High Throughput Sequencing Unit for providing us with the Chromium Controller (10 X Genomics) microfluidic partitioning instrument for single-cell encapsulation. The computational work for this article was partially performed on resources of the National Supercomputing Centre (NSCC), Singapore. This work was supported by funds from the National Medical Research Council Open Fund (MOH-000566 to KAK and GT and MOH-000645 to KAK), the Singapore Ministry of Education Academic Research Fund Tier 2 (MOE2019-T2-2-089 to KAK), NTU Research Scholarship to SYTL (predoctoral fellowship), and Nanyang President's Graduate Scholarship to FRT Parts of this work were also supported by the National Research Foundation and Ministry of Education Singapore under its Research Centre of Excellence Program (SCELSE).

## Additional information

### Funding

| Funder | Grant reference number | Author |
| --- | --- | --- |
| National Medical Research Council | MOH-000566 | Kimberly Kline Guillaume Thibault |

| Funder | Grant reference number | Author |
| --- | --- | --- |
| National Medical Research Council | MOH-000645 | Kimberly Kline |
| Ministry of Education - Singapore | MOE2019-T2-2-089 | Kimberly Kline |
| Nanyang Technological University | predoctoral fellowship | Stella Tue Ting Lee |
| Nanyang Technological University | Nanyang President's Graduate Scholarship | Frederick Reinhart Tanoto |
| National Research Foundation Singapore | Research Centre of Excellence Program (SCELSE) | Kimberly Kline |

The funders had no role in study design, data collection and interpretation, or the decision to submit the work for publication.

## Author contributions

Cenk Celik, Conceptualization, Resources, Data curation, Software, Formal analysis, Validation, Investigation, Visualization, Methodology, Writing – original draft, Writing – review and editing; Stella Tue Ting Lee, Investigation; Frederick Reinhart Tanoto, Formal analysis, Investigation, Methodology, Writing – original draft, Writing – review and editing; Mark Veleba, Resources, Investigation; Kimberly Kline, Conceptualization, Supervision, Funding acquisition, Methodology, Writing – original draft, Project administration, Writing – review and editing; Guillaume Thibault, Conceptualization, Supervision, Funding acquisition, Visualization, Methodology, Writing – original draft, Project administration, Writing – review and editing

## Author ORCIDs

Cenk Celik ![ORCID] https://orcid.org/0000-0001-8301-0172
Frederick Reinhart Tanoto ![ORCID] https://orcid.org/0000-0001-9289-4811
Mark Veleba ![ORCID] http://orcid.org/0000-0002-9516-8660
Guillaume Thibault ![ORCID] http://orcid.org/0000-0002-7926-4812

## Ethics

In vivo procedures were approved by the Animal Care and Use Committee of the Biological Resource Centre (Nanyang Technological University, Singapore) in accordance with the guidelines of the Agri-Food and Veterinary Authority and the National Advisory Committee for Laboratory Animal Research of Singapore (ARF SBS/NIEA-0314) and accordance with the approved institutional animal care and use committee (IACUC) protocols (#A19061) of Nanyang Technological University.

Reviewer #1 (Public Review): https://doi.org/10.7554/eLife.95113.3.sa1
Reviewer #2 (Public Review): https://doi.org/10.7554/eLife.95113.3.sa2
Author response https://doi.org/10.7554/eLife.95113.3.sa3

# Additional files

## Supplementary files

• Supplementary file 1. Number of cells in each Louvain cluster grouped by condition. Related to *Figure 1*.

• Supplementary file 2. Differential expression table of genes for the main class clusters. Related to *Figure 1C*.

• Supplementary file 3. Differential expression table of genes for the main uninfected class clusters. Related to *Figure 1C*.

• Supplementary file 4. Differential expression table of genes for the main infected class clusters. Related to *Figure 1C*.

• Supplementary file 5. Differential expression table of genes for sub-clustered keratinocytes. Related to *Figure 2*.

• Supplementary file 6. CellChat comparison for ligand:receptor pairs for the uninfected and infected datasets. Related to *Figures 2–6*.

• Supplementary file 7. Differential expression table of genes for sub-clustered fibroblasts. Related to *Figure 3*.

• Supplementary file 8. Differential expression table of genes for sub-clustered myeloid clusters. Related to *Figure 4*.

• Supplementary file 9. Differential expression table of genes for sub-clustered neutrophil clusters. Related to *Figure 5*.

• MDAR checklist

## Data availability

The accession number for the raw data reported in this paper is GSE229257. The processed data are available at Zenodo (https://doi.org/10.5281/zenodo.7608212).

The following dataset was generated:

| Author(s) | Year | Dataset title | Dataset URL | Database and Identifier |
|---|---|---|---|---|
| Celik C, Thibault G | 2023 | Gene expression profile at single cell level from untreated and *Enterococcus faecalis*-infected skin wounds | https://www.ncbi.nlm.nih.gov/geo/query/acc.cgi?acc=GSE229257 | NCBI Gene Expression Omnibus, GSE229257 |

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
