## [Editor Report · eLife assessment]

Wounds are commonly infected, which can lead to delayed or poor wound healing, thereby significantly impacting morbidity and overall quality of life for patients. This manuscript uses single cell RNA sequencing to try to understand the impact of infection on various cell types during wound healing in a mouse model. The methodology is **solid** and the results provide a **valuable** 'atlas' of the cellular changes associated with infected and uninfected wounds which will be of interest to the field.

---

## [Referee Report · Reviewer #1 (Public Review)]

Summary:

This is an interesting study that performs scRNA-Seq on infected and uninfected wounds. The authors sought to understand how infection with *E. faecalis* influences the transcriptional profile of healing wounds. The analysis demonstrated that there is a unique transcriptional profile in infected wounds with specific changes in macrophages, keratinocytes, and fibroblasts. They also speculated on potential crosstalk between macrophages and neutrophils and macrophages and endothelial cells using NicheNet analysis and CellChat. Overall the data suggest that infection causes keratinocytes to not fully transition which may impede their function in wound healing and that the infection greatly influenced the transcriptional profile of macrophages and how they interact with other cells.

Strengths:

It is a useful dataset to help to understand the impact of wound infection on transcription of specific cell types. The analysis is very thorough in terms of transcriptional analysis and uses a variety of techniques and metrics.

Weaknesses:

Some drawbacks of the study are the following. First the fact that it only has two mice per group, and only looks at one time point after wounding decreases the impact of the study. Wound healing is a dynamic and variable process so understanding the full course of the wound healing response would be very important to understand the impact of infection on the healing wound. The analysis has been bolstered by applying a cross-entropy test on the integrated dataset and to ensure robustness of the datasets (Fig S1F). Including unwounded skin in the scRNA-Seq would also lend a lot more significance to this study. However, this was technically challenging due to constraints with the number of immune cells in unwounded skin as described in the limitations section. Another drawback of the study is that mouse punch biopsies are very different than human wounds as they heal primarily by contraction instead of re-epithelialization like human wounds. The authors mitigated this somewhat be extracting the incisional parts of the wound. So while the conclusions are generally supported the scope of the work is somewhat limited.

---

## [Referee Report · Reviewer #2 (Public Review)]

Summary:

The authors have performed a detailed analysis of the complex transcriptional status of numerous cell types present in wounded tissue, including keratinocytes, fibroblasts, macrophages, neutrophils, and endothelial cells. The comparison between infected and uninfected wounds is interesting and the analysis suggests possible explanations for why infected wounds are delayed in their healing response.

Strengths:

The paper presents a thorough and detailed analysis of the scRNAseq data. The paper is clearly written and the conclusions drawn from the analysis are appropriately cautious. The results provide an important foundation for future work on the healing of infected and uninfected wounds.

Weaknesses:

The analysis is purely descriptive and no attempt is made to validate whether any of the factors identified are playing functional roles in wound healing. Such experiments would be appropriate for followup work. The experimental setup is analyzing a single time point and does not include a comparison to unwounded skin. Nevertheless, the present data do provide a useful point of comparison for the field.

---

## [Author Response]

The following is the authors’ response to the original reviews.

**Public Reviews:**

**Reviewer #1 (Public Review):**
Summary:This is an interesting study that performs scRNA-Seq on infected and uninfected wounds. The authors sought to understand how infection with *E. faecalis* influences the transcriptional profile of healing wounds. The analysis demonstrated that there is a unique transcriptional profile in infected wounds with specific changes in macrophages, keratinocytes, and fibroblasts. They also speculated on potential crosstalk between macrophages and neutrophils and macrophages and endothelial cells using NicheNet analysis and CellChat. Overall the data suggest that infection causes keratinocytes to not fully transition which may impede their function in wound healing and that the infection greatly influenced the transcriptional profile of macrophages and how they interact with other cells.Strengths:It is a useful dataset to help understand the impact of wound infection on the transcription of specific cell types. The analysis is very thorough in terms of transcriptional analysis and uses a variety of techniques and metrics.Weaknesses:Some drawbacks of the study are the following. First, the fact that it only has two mice per group, and only looks at one time point after wounding decreases the impact of the study. Wound healing is a dynamic and variable process so understanding the full course of the wound healing response would be very important to understand the impact of infection on the healing wound. Including unwounded skin in the scRNA-Seq would also lend a lot more significance to this study. Another drawback of the study is that mouse punch biopsies are very different than human wounds as they heal primarily by contraction instead of reepithelialization like human wounds. So while the conclusions are generally supported the scope of the work is limited.

Thank you for your thoughtful review and acknowledgment of the thoroughness of our analysis.

First, the fact that it only has two mice per group, and only looks at one time point after wounding decreases the impact of the study.

We acknowledge your concerns regarding the limitations of our study, particularly regarding the small number of mice per group and the examination of only one time point post-wounding. We agree that a more comprehensive analysis across multiple time points would provide a deeper understanding of the temporal changes induced by infection. While our primary focus in this study was to elucidate the foundational responses to bacteria-infected wounds, we attempted to augment our analysis by incorporating publicly available datasets of similar nature. However, these datasets lacked power in terms of cell number and populations. Nonetheless, we have bolstered our analysis by applying a crossentropy test on the integrated dataset and reporting its significance (Figure S1F), ensuring the robustness of our single-cell RNA sequencing datasets.

Including unwounded skin in the scRNA-Seq would also lend a lot more significance to this study.

We also recognize the significance of comparing infected wounds to unwounded skin to establish a baseline for transcriptional changes. While we attempted to incorporate publicly available unwounded skin samples into our analysis, we encountered limitations in the number of cells, particularly within the immune population. This constraint is addressed in the Limitations section of the manuscript.

Another drawback of the study is that mouse punch biopsies are very different than human wounds as they heal primarily by contraction instead of re-epithelialization like human wounds.

Regarding the concern about differences between murine and human wound healing mechanisms, we took measures during tissue isolation to mitigate this issue, extracting incisions of the wounds rather than contracted tissues. Despite the primary mode of wound closure in mice being contraction, we believe our analysis still offers valuable insights into cellular responses to infection relevant to human wound healing.

We appreciate your constructive criticism of our study. Despite these constraints, we believe our work provides valuable insights into the transcriptional changes induced by infection in healing wounds.

**Reviewer #2 (Public Review):**
Summary:The authors have performed a detailed analysis of the complex transcriptional status of numerous cell types present in wounded tissue, including keratinocytes, fibroblasts, macrophages, neutrophils, and endothelial cells. The comparison between infected and uninfected wounds is interesting and the analysis suggests possible explanations for why infected wounds are delayed in their healing response.Strengths:The paper presents a thorough and detailed analysis of the scRNAseq data. The paper is clearly written and the conclusions drawn from the analysis are appropriately cautious. The results provide an important foundation for future work on the healing of infected and uninfected wounds.Weaknesses:The analysis is purely descriptive and no attempt is made to validate whether any of the factors identified are playing functional roles in wound healing. The experimental setup is analyzing a single time point and does not include a comparison to unwounded skin.

We are thankful for your acknowledgment of the thoroughness of our analysis and the cautious nature of our conclusions.

The analysis is purely descriptive, and no attempt is made to validate whether any of the factors identified are playing functional roles in wound healing.

Regarding your concern about the purely descriptive nature of our analysis and the lack of functional validation of identified factors, we agree on the importance of understanding the functional roles of transcriptional changes in wound healing. To address this limitation, we plan to conduct functional experiments, such as perturbation assays or in vivo validation studies, to validate the roles of specific factors identified in our analysis.

The experimental setup is analyzing a single time point and does not include a comparison to unwounded skin.

We acknowledge the importance of comparing wounded tissue to unwounded skin to establish a baseline for understanding transcriptional changes. This point is noted and acknowledged in the limitations section of our manuscript.

We appreciate your feedback and assure you that we will consider your suggestions in future iterations of our research.

**Recommendations For The Authors:**

We are grateful for the positive overall assessment of our revised work by the reviewers. Critical comments on specific aspects of our work are listed verbatim below followed by our responses.

**Reviewer 1 (Recommendations for the Authors):**
(1) The figures are a bit cluttered and hard to parse out. The different parts of the figure seem to be scattered all over the place with no consistent order.

Thank you for your feedback regarding the figures in our manuscript. We acknowledge your concern that some panels may appear cluttered and challenging to navigate. In response, we made concerted efforts to declutter certain panels, taking into account page size constraints and ensuring a minimum font size for readability.

(2) I didn't really understand what the last sentence on page 6 meant. Is this meant to say that these could be biomarkers of infection?

We thank the reviewer for noting this lack of clarity. We revised the statement.

Updated manuscript (lines 111-113)

“Overall, the persistent *E. faecalis* infection contributed to higher Tgfb1 expression, whilst Pdgfa levels remained low, correlating with delayed wound healing.”

(3) >(3) A reference on page 19 didn't format correctly.

We thank the reviewer for catching the typos. We corrected the reference formatting.

Updated manuscript (lines 503-505)

“We confirm the immune-suppressive role of *E. faecalis* in wound healing, consistent with previous findings in different experimental settings (Chong et al., 2017; Kao et al., 2023; Tien et al., 2017).”

(4) The title doesn't really address the scope of the finding which goes beyond immunomodulatory.

The reviewer is correct! We therefore revised the title to cover all aspects of the study as:

“Decoding the complexity of delayed wound healing following *Enterococcus faecalis* infection”

**Reviewer 2 (Recommendations for the Authors):**
(1) On page 6, the expression of Tgfb1 is described as "aggravated" by wounding alone. I am not sure whether this means Tgfb1 levels are increased or decreased. It appears from the data that it is increased, which was confusing to me since I interpreted "aggravated" as meaning decreased. So perhaps a different more straightforward word could be used to describe the data.

We modified this ambiguous statement to:

Updated manuscript (lines 105-106)

“By contrast, wounding alone resulted in higher transforming growth factor beta 1 (Tgfb1) expression.”

(2) On page 7, the authors state that "cells from infected wounds...demonstrated distinct clustering patterns compared to cells from uninfected wounds (Figure S1F)" but when I look at the data in this figure, I cannot really see a difference. Perhaps the differences could be more clearly highlighted?

Thank you for pointing out this issue. We appreciate the reviewer's comment. We utilized the crossentropy test for statistical comparison, employing UMAP embedding space data. While the data underwent batch correction based on infection status, the UMAP plots for each condition may appear visually similar. However, it's important to note that the number of cells per clusters between the infected and uninfected conditions varies significantly. This aspect influences the selection of points (cells) and their nearest neighbours for statistical testing within each cluster in the embedding space. To address this concern, we have included a table indicating the number of cells per cell type alongside the plot (Figure S1F), providing additional context for the interpretation of our results.

**Author response table 1. sa3table1:** 

	Uninfected	Infected
Basal	861	860
Dendritic cells	169	203
Endothelial cells	469	1,246
Fibroblast	1,700	2,699
Macrophage	1,912	2,255
Memory T cell	97	68
Neuron	152	101
Neutrophil	270	1,170
Outer bulge	336	364
Sebaceous gland	1,184	1,637
Suprabasal	1,886	2,557
Hair follicle suprabasal	283	336
Vascular smooth muscle	130 _	159 _

(3) On page 8, Zeb2hi cells are described as "immunosuppressive" and yet the genes are highlighted to express in include Cxcl2 and IL1b which I would classify as inflammatory, not immunosuppressive. Can the authors be a bit more clear on why they describe the phenotype of these cells as "immunosuppressive"?

We agree with the reviewer that this is a bit counterintuitive. Conventionally, CXCL2 is thought to be chemoattractant for neutrophil recruitment. However, the infection-specific keratinocyte cluster expressing Cxcl2, Il1b, Wfdc17 along with Zeb2 and Thbs1 indicate their myeloid-derived suppressor cell-like features, which play immunosuppressive roles during infection and in cancer (Alshetaiwi et al., 2020; Siriwach et al., 2022; Veglia et al., 2021).

Updated manuscript (lines 159-163)

“As the barrier to pathogens, keratinocytes secrete a broad range of cytokines that can induce inflammatory responses (Alshetaiwi et al., 2020; Siriwach et al., 2022; Veglia et al., 2021). However, Zeb2hi keratinocytes co-expressing Cxcl2, Il1b, and Wfdc17, indicate myeloidderived suppressor cell-like phenotype which implies an immunosuppressive environment (Hofer et al., 2021; Veglia et al., 2021).”

(4) On pages 8-9, Keratinocytes are described to express MHC class II. I find this quite unexpected since class II is usually thought to be expressed primarily by APCs such as DCs and B cells. Is there a precedent for keratinocytes to express class II? The authors should acknowledge that this is unexpected and in need of further validation, or support the claim with references in which class II expression has been previously observed on keratinocytes (and is thus not unexpected)

Although MHC class II expression is predominantly on immune cells, an antigen-presenting role for keratinocytes has been reported in many studies (Banerjee et al., 2004; Black et al., 2007; Carr et al., 1986; Gawkrodger et al., 1987; Jiang et al., 2020; Li et al., 2022; Oh et al., 2019; Tamoutounour et al., 2019). Therefore, antigen-presenting role of keratinocytes is known and expected, and we think that this should be further investigated in in the context of wound infection.

Updated manuscript (lines 177-179)

“These genes are associated with the major histocompatibility complex (MHC) class II, suggesting a self-antigen presenting keratinocyte population, which have a role in costimulation of T cell responses (Meister et al., 2015; Tamoutounour et al., 2019).”

REFERENCES

Alshetaiwi, H., Pervolarakis, N., McIntyre, L. L., Ma, D., Nguyen, Q., Rath, J. A., Nee, K., Hernandez, G., Evans, K., Torosian, L., Silva, A., Walsh, C., & Kessenbrock, K. (2020). Defining the emergence of myeloid-derived suppressor cells in breast cancer using single-cell transcriptomics. Science Immunology, 5(44), eaay6017. https://doi.org/10.1126/sciimmunol.aay6017

Banerjee, G., Damodaran, A., Devi, N., Dharmalingam, K., & Raman, G. (2004). Role of keratinocytes in antigen presentation and polarization of human T lymphocytes. Scandinavian Journal of Immunology, 59(4), 385–394. https://doi.org/10.1111/j.0300-9475.2004.01394.x

Black, A. P. B., Ardern-Jones, M. R., Kasprowicz, V., Bowness, P., Jones, L., Bailey, A. S., & Ogg, G. S. (2007). Human keratinocyte induction of rapid effector function in antigen-specific memory CD4+ and CD8+ T cells. European Journal of Immunology, 37(6), 1485–1493. https://doi.org/10.1002/eji.200636915

Carr, M. M., McVittie, E., Guy, K., Gawkrodger, D. J., & Hunter, J. A. (1986). MHC class II antigen expression in normal human epidermis. Immunology, 59(2), 223–227.

Gawkrodger, D. J., Carr, M. M., McVittie, E., Guy, K., & Hunter, J. A. (1987). Keratinocyte expression of MHC class II antigens in allergic sensitization and challenge reactions and in irritant contact dermatitis. The Journal of Investigative Dermatology, 88(1), 11–16. https://doi.org/10.1111/1523-1747.ep12464641

Jiang, Y., Tsoi, L. C., Billi, A. C., Ward, N. L., Harms, P. W., Zeng, C., Maverakis, E., Kahlenberg, J. M., & Gudjonsson, J. E. (2020). Cytokinocytes: The diverse contribution of keratinocytes to immune responses in skin. JCI Insight, 5(20), e142067, 142067. https://doi.org/10.1172/jci.insight.142067

Li, D., Cheng, S., Pei, Y., Sommar, P., Kärner, J., Herter, E. K., Toma, M. A., Zhang, L., Pham, K., Cheung, Y. T., Liu, Z., Chen, X., Eidsmo, L., Deng, Q., & Xu Landén, N. (2022). Single-Cell Analysis Reveals Major Histocompatibility Complex II‒Expressing Keratinocytes in Pressure Ulcers with Worse Healing Outcomes. The Journal of Investigative Dermatology, 142(3 Pt A), 705–716. https://doi.org/10.1016/j.jid.2021.07.176

Oh, S., Chung, H., Chang, S., Lee, S.-H., Seok, S. H., & Lee, H. (2019). Effect of Mechanical Stretch on the DNCB-induced Proinflammatory Cytokine Secretion in Human Keratinocytes. Scientific Reports, 9(1), 5156. https://doi.org/10.1038/s41598-019-41480-y

Siriwach, R., Ngo, A. Q., Higuchi, M., Arima, K., Sakamoto, S., Watanabe, A., Narumiya, S., & Thumkeo, D. (2022). Single-cell RNA sequencing identifies a migratory keratinocyte subpopulation expressing THBS1 in epidermal wound healing. iScience, 25(4), 104130. https://doi.org/10.1016/j.isci.2022.104130

Tamoutounour, S., Han, S.-J., Deckers, J., Constantinides, M. G., Hurabielle, C., Harrison, O. J., Bouladoux, N., Linehan, J. L., Link, V. M., Vujkovic-Cvijin, I., Perez-Chaparro, P. J., Rosshart, S. P., Rehermann, B., Lazarevic, V., & Belkaid, Y. (2019). Keratinocyte-intrinsic MHCII expression controls microbiota-induced Th1 cell responses. Proceedings of the National Academy of Sciences of the United States of America, 116(47), 23643–23652. https://doi.org/10.1073/pnas.1912432116

Veglia, F., Sanseviero, E., & Gabrilovich, D. I. (2021). Myeloid-derived suppressor cells in the era of increasing myeloid cell diversity. Nature Reviews. Immunology, 21(8), 485–498. https://doi.org/10.1038/s41577-020-00490-y